



# Effects of Strongly Enhanced Atmospheric Methane Concentrations in a Fully Coupled Chemistry-Climate Model

Laura Stecher[1], Franziska Winterstein[1], Martin Dameris[1], Patrick Jöckel[1], Michael Ponater[1], and Markus Kunze[2]

[1]Deutsches Zentrum für Luft- und Raumfahrt (DLR), Institut für Physik der Atmosphäre, Oberpfaffenhofen, Germany
[2]Freie Universität Berlin, Berlin, Germany

**Correspondence:** Laura Stecher (Laura.Stecher@dlr.de)

**Abstract.** In a previous study the quasi-instantaneous chemical impacts (rapid adjustments) of strongly enhanced methane ($CH_4$) mixing ratios have been analyzed. However, to quantify the influence of the respective slow climate feedbacks on the chemical composition it is necessary to include the radiation driven temperature feedback. Therefore, we perform sensitivity simulations with doubled and fivefold present-day (year 2010) $CH_4$ mixing ratios with the chemistry-climate model EMAC

and include in a novel set-up a mixed layer ocean model to account for tropospheric warming.

We find that the slow climate feedbacks counteract the reduction of the hydroxyl radical in the troposphere, which is caused by the strongly enhanced $CH_4$ mixing ratios. Thereby also the resulting prolongation of the tropospheric $CH_4$ lifetime is weakened compared to the quasi-instantaneous response considered previously.

Changes in the stratospheric circulation evolve clearly with the warming of the troposphere. The Brewer-Dobson circulation

strengthens, affecting the response of trace gases, such as ozone, water vapour and $CH_4$ in the stratosphere, and also causing stratospheric temperature changes. In the middle and upper stratosphere, the increase of stratospheric water vapour is reduced with respect to the quasi-instantaneous response. Weaker increases of the hydroxyl radical cause the chemical depletion of $CH_4$ to be less strongly enhanced and thus the in situ source of stratospheric water vapour as well. However, in the lower stratosphere water vapour increases more strongly when tropospheric warming is accounted for enlarging its overall radiative impact.

The response of the stratospheric adjusted temperatures driven by slow climate feedbacks is dominated by these increases of stratospheric water vapour, as well as strongly decreased ozone mixing ratios above the tropical tropopause, which result from enhanced tropical upwelling.

While rapid radiative adjustments from ozone and stratospheric water vapour make an essential contribution to the effective $CH_4$ radiative forcing, the radiative impact of the respective slow feedbacks is rather moderate. In line with this, the climate

sensitivity from $CH_4$ changes in this chemistry-climate model setup is not significantly different from the climate sensitivity in carbon dioxide-driven simulations, provided that the $CH_4$ effective radiative forcing includes the rapid adjustments from ozone and stratospheric water vapour changes.



# 1 Introduction

Methane ($CH_4$) is the second most important anthropogenically influenced greenhouse gas (GHG). Apart from its direct radiative impact (RI), $CH_4$ is chemically active and induces chemical feedbacks relevant for climate and air quality. Through its most important tropospheric sink, the oxidation with the hydroxyl radical (OH), it affects the oxidation capacity of the atmosphere and thus its own lifetime (e.g., Saunois et al., 2016b; Voulgarakis et al., 2013; Winterstein et al., 2019). $CH_4$ oxidation is further an important source of stratospheric water vapour (SWV) (e.g., Frank et al., 2018) and affects the ozone ($O_3$) concentration in

troposphere and stratosphere via secondary feedbacks. Chemical feedbacks from $O_3$ and SWV contribute significantly to the total RI induced by $CH_4$ (e.g., Fig. 8.17 in IPCC, 2013; Winterstein et al., 2019). The abundance of $CH_4$ in the atmosphere is rising rapidly at present (e.g., Nisbet et al., 2019). Furthermore, emissions from natural $CH_4$ sources can be prone to climate change and have the potential to strongly enhance atmospheric $CH_4$ concentrations (Dean et al., 2018). Together with its relevance as a GHG, the latter underlines the importance of examining implications of strongly increased $CH_4$ abundances in the

atmosphere.

Chemistry-climate models (CCMs) are useful tools for such studies. A CCM is a General Circulation model (GCM) that is interactively coupled to a comprehensive chemistry module. This online two-way coupling is necessary to assess, on the one hand, chemically induced changes of radiatively active gases and their feedback on temperature, and on the other hand feedbacks on chemical processes driven by changes of the climatic state (e.g. temperature, circulation or precipitation). A range

of CCM studies analysed the sensitivity of other atmospheric constituents, such as $O_3$ (Kirner et al., 2015; Morgenstern et al., 2018), SWV (Revell et al., 2016) and OH and $CH_4$ lifetime (Voulgarakis et al., 2013), to different projections of $CH_4$ mixing ratios. However, these studies did not focus on the climate impact of $CH_4$. Other recent studies assessing climate feedbacks and climate sensitivity of $CH_4$ did not include radiative contributions from chemical feedbacks in their analysis (Modak et al., 2018; Smith et al., 2018; Richardson et al., 2019).

Winterstein et al. (2019) assessed chemical feedback processes and their RI in sensitivity simulations forced by 2-fold ($2\times$) and 5-fold ($5\times$) present-day (year 2010) $CH_4$ mixing ratios. As their simulation set-up prescribed sea surface temperatures (SSTs) and sea ice concentrations (SICs) and thus suppressed surface temperature changes, the parameter changes in their simulations have the character of rapid adjustments (e.g., Forster et al., 2016; Smith et al., 2018). In the effective radiative forcing (ERF) framework, rapid adjustments of radiatively active species are counted as part of the forcing and are to be distinguished

from slow climate feedbacks that are coupled to surface temperature changes (Sherwood et al., 2015). Climate sensitivity parameters, reflecting the degree of surface temperature change per unit forcing, have been found to be less dependent on the forcing agent with this definition compared to previous definitions of radiative forcing (RF) (e.g., Shine et al., 2003; Hansen et al., 2005; Richardson et al., 2019).

As a follow-up on Winterstein et al. (2019), we assess the respective SST-driven climate feedbacks, their effect on the quasi-

instantaneous response of the chemical composition, and consequently resulting radiative feedbacks. Consistent with Winterstein et al. (2019), we perform sensitivity simulations with $2\times$ and $5\times$ present-day $CH_4$ mixing ratios with the ECHAM/MESSy Atmospheric Chemistry (EMAC) CCM (Jöckel et al., 2016), but this time coupled to a mixed layer ocean (MLO) model in-





stead of prescribing SSTs and SICs. To our knowledge, this is the first study assessing the response to strong increases of $CH_4$ mixing ratios in a fully coupled CCM, meaning that the interactive model system includes atmospheric dynamics, atmospheric

chemistry, and ocean thermodynamics.

Our simulation strategy is explained in Sect. 2. The discussion of results in Sect. 3 starts with a brief evaluation of the reference $CH_4$ mixing ratio against observations and an assessment of the MLO model (Sect. 3.1), followed by the analyses of tropospheric warming and associated climate feedbacks in the MLO simulations (Sect. 3.2). In Sect. 3.3 we assess implications of SST-driven climate feedbacks on the chemical composition of the atmosphere in comparison to the quasi-instantaneous

response and quantify the resulting radiative feedbacks and the climate sensitivity. We further discuss contributions from feedbacks of radiatively active gases and from circulation changes to the stratospheric temperature response. In Sect. 4 we summarize our conclusions and give a brief outlook.

## 2 Description of the model and simulation strategy

We use the CCM ECHAM/MESSy Atmospheric Chemistry (EMAC; Jöckel et al., 2016) for this study. Following on from the

sensitivity simulations with prescribed SSTs and SICs that were analysed by Winterstein et al. (2019), we performed a second set of sensitivity simulations with the MESSy submodel MLOCEAN (Kunze et al. (2014); original code by Roeckner et al. (1995)) coupled to EMAC. The set-up of the MLO simulations is designed to follow the set-up of the simulations described by Winterstein et al. (2019) closely. We conducted all simulations at a resolution of T42L90MA, corresponding to a quadratic Gaussian grid of approximately $2.8° \times 2.8°$ resolution in latitude and longitude and 90 levels with the uppermost level centered

around 0.01 hPa in the vertical.

According to the simulation concept of Winterstein et al. (2019), we performed one reference simulation (REF MLO) and two sensitivity simulations (S2 MLO and S5 MLO) including the MLO model, all as equilibrium climate simulations. The simulations with prescribed SSTs and SICs are denoted REF fSST, S2 fSST and S5 fSST here. All simulations considered for the analysis are listed in Tab. 1. The MLO simulations have been performed with a more recent version of the Modular

Earth Submodel System (MESSy; 2.54.0 instead of 2.52). The updates include changes in the chemistry module Module Efficiently Calculating the Chemistry of the Atmosphere (MECCA; Sander et al. (2011)) that are discussed in Appendix A. However, inherent differences between the MLO and fSST simulations do not directly distort the evaluation, as the differences between response signals relative to the respective reference simulations, and not the direct differences between the sensitivity simulations, are analysed.

A spin-up phase of at least ten years is excluded from the analysis of each simulation to provide quasi-steady-state conditions. S2 MLO and S5 MLO were initialized from the spun-up state of REF MLO and spun-up over a 10-year period, followed by a 20-year equilibrium used for the analysis. We chose to simulate a 30-year equilibrium for the analysis of REF MLO after S2 MLO and S5 MLO branched off, so that the complete 20 years used for the analysis of S2 MLO and S5 MLO are covered by this simulation as well.





**Table 1.** Overview of the two sets of sensitivity simulations (fSST and MLO) with one reference simulation and two sensitivity simulations. The simulations with prescribed SSTs and SICs have already been analysed by Winterstein et al. (2019). The simulation REF QFLX is used to determine the heat flux correction for the simulations including the MLO model.

| Simulation | $CH_4$ lower boundary | SSTs, SICs | MESSy version |
|---|---|---|---|
| REF fSST | 1.8 ppmv | | |
| S2 fSST | $2 \times$ REF fSST | prescribed (Rayner et al., 2003) | 2.52 |
| S5 fSST | $5 \times$ REF fSST | | |
| REF MLO | 1.8 ppmv | mixed layer ocean (MLO) | |
| S2 MLO | $2 \times$ REF MLO | MESSy submodel MLOCEAN | 2.54.0 |
| S5 MLO | $5 \times$ REF MLO | | |
| REF QFLX | 1.8 ppmv | prescribed (Rayner et al., 2003) | d2.53.0.26 |

The MLO simulations have been initialized with the equilibrium $CH_4$ fields of the respective fSST simulations, thus the initial $CH_4$ fields of S2 MLO and S5 MLO were implicitly scaled by two and five, respectively. Alike the fSST simulations, the $CH_4$ lower boundary mixing ratios of the MLO simulations are prescribed by Newtonian relaxation (i.e. nudging). The lower boundary $CH_4$ mixing ratios of REF MLO are nudged to the same reference as REF fSST, namely a zonal mean observation based estimate of the year 2010 from marine boundary layer sites. The observational data are provided by the Advanced

Global Atmospheric Gases Experiment (AGAGE; http://agage.mit.edu/) and the National Oceanic and Atmospheric Administration/Earth System Research Laboratory (NOAA/ESRL; https://www.esrl.noaa.gov/). The lower boundary $CH_4$ mixing ratios of S2 and S5 are nudged towards the $2 \times$ and the $5 \times$ of this reference, respectively. The resulting global mean lower boundary $CH_4$ mixing ratio is about 1.8 parts per million volume (ppmv) for both reference simulations, 3.6 ppmv for both doubling, and 9.0 ppmv for both fivefolding experiments. All other prescribed boundary conditions, such as emission fluxes, in the sensitivity

simulations are identical to the respective reference simulations and represent conditions of the year 2010 in general.

   In the MLO simulations, the SSTs, the ice thicknesses, and the ice temperatures at ocean gridpoints are calculated by the MESSy submodel MLOCEAN. A MLO model accounts for the ocean's heat capacity without simulating the oceanic circulation explicitly. To simulate realistic SSTs with the MLO, a heat flux correction term needs to be added to the surface energy balance. We derived a monthly climatology of this heat flux correction from a control simulation with prescribed SSTs

and SICs, named REF QFLX. REF QFLX uses the same monthly climatology of SSTs and SICs that was used for the fSST simulations, i. e. a monthly climatology representing the years 2000 to 2009 based on global analyses of the HadISST1 data set (Rayner et al., 2003).

   In the following, the response to increased $CH_4$ in the MLO simulations is assessed as the difference of S2 MLO and S5 MLO with respect to REF MLO. The effects of SST-driven climate feedbacks are identified as the difference between responses in the

MLO and fSST simulations. The RIs induced by changes of individual radiatively active gases are assessed using the EMAC option for multiple radiation calls in the submodel RAD (Dietmüller et al., 2016), as explained in more detail by Winterstein





et al. (2019). The first radiation call receives the reference mixing ratios of all chemical species, i.e. $CH_4$, $O_3$ and water vapour ($H_2O$). In the following radiation calls each of the species individually, and all combined, are exchanged by climatological means derived from the sensitivity simulations (S2 and S5). From these perturbed radiation fluxes the stratospheric-adjusted

RI is calculated (Stuber et al., 2001; Dietmüller et al., 2016).

## 3 Discussion of results

### 3.1 Assessment of reference simulations

The simulation set-up of the reference simulation, REF MLO, aims to represent conditions typical for the year 2010. For a detailed assessment and evaluation of EMAC in general, we refer to Jöckel et al. (2016). We have evaluated the REF MLO

$CH_4$ mixing ratios to ensure that the latter represent conditions of 2010 sufficiently realistic. The REF MLO $CH_4$ mixing ratios were compared to three different observational data sets that are independent from the observational estimate that serves as input for the lower boundary condition to ensure an objective evaluation. These are balloon-borne measurements conducted in the period from 1992 to 2006 from Röckmann et al. (2011), observations of a portable Fourier transform spectrometer onboard the research vessel Polarstern during a cruise from Cape Town to Bremerhaven on the Atlantic in 2014 (Klappenbach et al.,

2015) and observations from the Total Carbon Column Observing Network (TCCON; Wunch et al., 2011) from the period 2009 to 2014. The vertical profile, the north-south gradient and the annual cycle of REF MLO $CH_4$ generally agree well with the corresponding data (not shown). Consistent with REF fSST (see Winterstein et al., 2019), there is a negative bias between the REF MLO and the observed total $CH_4$ columns of less than 4 % (not shown). Given that relative comparisons between sensitivity simulations and the reference are the main target of our analysis, REF MLO represents $CH_4$ conditions of the year

2010 sufficiently realistic for our purpose.

Since this study is one of the first to use the MLOCEAN submodel in MESSy, we have carefully checked whether REF MLO reproduces SSTs and SICs of the climatology that was used to determine the heat flux correction with sufficient accuracy. The spatial pattern of the SST climatology is realistically reproduced in REF MLO (see Fig. S1). The largest differences are found at higher latitudes, where a reduction in sea ice area leads to higher SSTs, as exposed sea water is warmer than sea ice. REF MLO

underestimates the monthly climatology of sea ice area in the Southern Hemisphere (SH) in all seasons, except for austral summer (see Fig. S2). The reduction of SIC results in up to 1.5 K higher SSTs in the Southern Ocean in REF MLO compared to the prescribed climatology (see Fig. S1). Zonal mean air temperatures in the SH extra-tropical troposphere are likewise up to 1 K higher in REF MLO compared to REF QFLX on annual average (not shown). As the contribution of Antarctic sea ice melting to global surface albedo feedback and climate response is comparatively small, a substantial underestimation of the

climate sensitivity from this effect is not to be expected.

In the Northern Hemisphere (NH), the monthly climatology of sea ice area is generally well reproduced (see Fig. S2). However, in boreal winter and spring REF MLO overestimates the prescribed climatology of sea ice area with a maximum deviation of $1.33 \times 10^9$ km$^2$ in April. The larger SICs result in about 0.5 K lower SSTs on annual average in REF MLO in the Greenland Sea and in the Barents Sea (see Fig. S1), where the increase of SIC is located (not shown). In the Hudson Bay





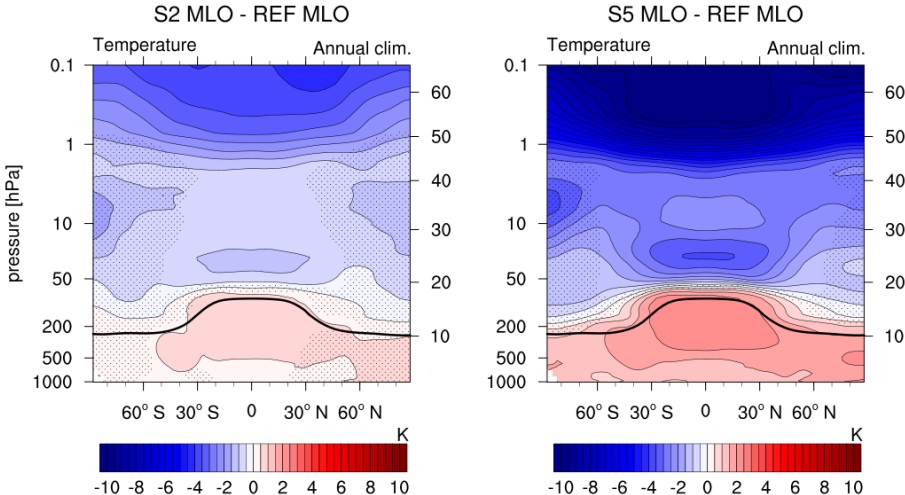

**Figure 1.** Absolute annual zonal mean temperature differences between the sensitivity simulations S2 MLO (left) and S5 MLO (right) and REF MLO in K. Non-stippled areas are significant on the 95 % confidence level according to a two sided Welch's test. The solid black line indicates the climatological tropopause height of REF MLO.

and in the Labrador Sea, on the other hand, the sea ice cover is reduced in REF MLO resulting in about 1 K higher SSTs in REF MLO compared to the prescribed climatology (see Fig. S1). The deviation from the prescribed climatology is strongest in this region in boreal summer. In summary, REF MLO simulates sufficiently realistic oceanic conditions for our purpose.

### 3.2 Tropospheric temperature response and associated climate feedbacks

The tropospheric temperature response to enhanced $CH_4$ mixing ratios can freely develop in the MLO sensitivity simulations (see Fig. 1). The temperature change patterns of S2 MLO and S5 MLO show the expected warming of the troposphere and cooling of the stratosphere (e.g., IPCC, 2013). The stratospheric cooling is less pronounced than in carbon dioxide ($CO_2$)-driven climate change simulations, since the $CH_4$ cooling is mainly caused by associated $O_3$ and $H_2O$ adjustments (Kirner et al., 2015; Winterstein et al., 2019). Maximum warming in polar regions and in the upper tropical troposphere is also consistent with changes expected from increased levels of GHGs (e.g., Chap. 12 in IPCC, 2013). $CH_4$ doubling (fivefolding) leads to temperature increases of up to 1 K (3 K) in the Arctic on annual average. Antarctica also warms up particularly strongly in the S5 MLO scenario with a maximum warming of up to 3 K. As a result of the especially strong warming in polar regions, the sea ice area is reduced in both sensitivity simulations with respect to the reference (compare Fig. S2).

The Brewer-Dobson circulation (BDC) is expected to accelerate in a warming climate (Rind et al., 1990; Butchart and Scaife, 2001; Garcia and Randel, 2008; Butchart, 2014; Eichinger et al., 2019). Feedbacks on the chemical composition of the atmosphere, especially of the stratosphere, which result from changes of the BDC are of particular interest in this study, as they will modify the mainly chemically induced changes discussed by Winterstein et al. (2019). The BDC influences the





spatial distribution of trace gases, such as $O_3$, $H_2O$, and $CH_4$, in the stratosphere and also their transport from the troposphere into the stratosphere (Butchart, 2014). In Fig. 2 we examine the response of the residual mean streamfunction to quantify changes of the BDC. There is indeed a strengthening of the residual mean circulation in both, S2 MLO and S5 MLO, with

respect to REF MLO and it is detected in both hemispheres. The change of the residual mean streamfunction is stronger and extends to higher altitudes for the simulation S5 MLO, but the annual mean patterns are consistent in both MLO sensitivity simulations. The maximum change of about $0.7 \times 10^9$ kg s$^{-1}$ for S5 MLO is located at about 100 hPa. Upward motion is increased in the tropics, which is balanced by an increase of downwelling between $30°$–$60°$ latitude in both hemispheres. The change of the residual mean streamfunction is stronger and reaches higher in the respective winter hemisphere in S5 MLO (see

Fig. S3 and Fig. S5). The BDC response in the MLO simulations is considerably stronger than in the respective fSST sensitivity simulations. This is expected, since the main driver of changes in the BDC is tropospheric warming (Butchart, 2014). We note that changes of the residual mean streamfunction below the tropical tropopause in response to $CH_4$ increase exhibit different patterns in the fSST and MLO simulations (see Fig. 2). A similar feature has been noticed and discussed in $CO_2$ increase simulations, too (e.g. Bony et al., 2013). However, trying to explain the origin of these tropospheric differences would leave

the scope of the present paper, which focuses on stratospheric trace gas feedbacks to $CH_4$ increase. The latter are influenced by the more distinct strengthening of the BDC in the MLO experiments, as we will show in the next section.

### 3.3  Influence of interactive SSTs

#### 3.3.1  Chemical composition

Winterstein et al. (2019) analysed the quasi-instantaneous impact of doubled and fivefold $CH_4$ mixing ratios on the chemical
composition of the atmosphere. In this section we investigate how tropospheric warming and associated climate feedbacks (see Sect. 3.2) modify these rapid adjustment patterns. For this purpose the difference patterns of the MLO sensitivity simulations are compared to those of the fSST simulations.

**Tropospheric $CH_4$ lifetime and OH**

Winterstein et al. (2019) found a near-linear prolongation of the tropospheric $CH_4$ lifetime, related to the oxidation with OH,
with increasing scaling factor of the $CH_4$ mixing ratio. The OH-oxidation is the most important sink of $CH_4$ in the troposphere (e.g., Saunois et al., 2016a). The amount of oxidised $CH_4$ affects the OH mixing ratio as the reaction consumes OH, which in turn feeds back on the atmospheric $CH_4$ lifetime. In this study, consistent with Winterstein et al. (2019), the $CH_4$ lifetime is calculated according to Jöckel et al. (2016) as

$$\tau_{CH_4} = \frac{\sum\limits_{b \in B} m_{CH_4}}{\sum\limits_{b \in B} k_{CH_4+OH}(T) \cdot c_{air}(T,p,q) \cdot x_{OH} \cdot m_{CH_4}}, \tag{1}$$

with $m_{CH_4}$ being the mass of $CH_4$ in kg, $k_{CH_4+OH}(T)$ the temperature dependent reaction rate coefficient of the reaction $CH_4 + OH \rightarrow$ products in [cm$^3$ s$^{-1}$], $c_{air}$ the concentration of air in [mol cm$^{-3}$] and $x_{OH}$ the mole fraction of OH in



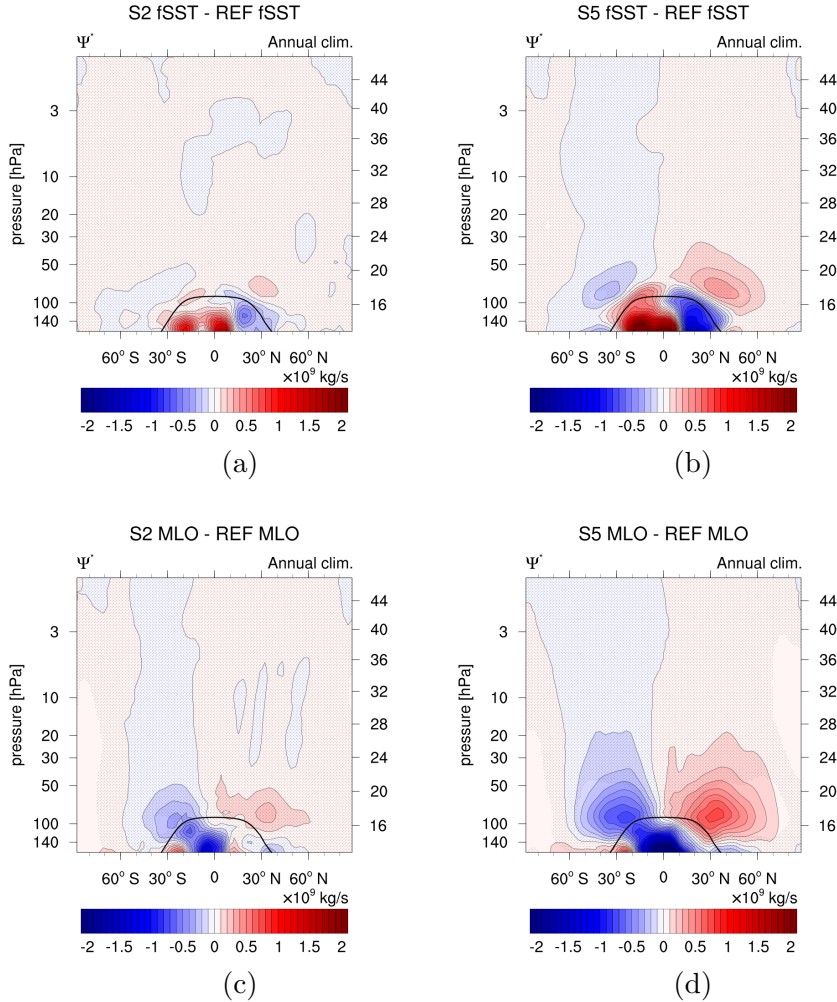

**Figure 2.** Absolute differences of the annual zonal mean residual streamfunction between the sensitivity simulations (a) S2 fSST, (b) S5 fSST, (c) S2 MLO, (d) S5 MLO compared to their respective reference in $10^9$ kg s$^{-1}$. Non-stippled areas are significant on the 95 % confidence level according to a two sided Welch's test. The solid black line indicates the climatological tropopause height of REF MLO.

[mol mol$^{-1}$] in all grid boxes b $\in$ B. B is the region, for which the lifetime should be calculated, e.g. all grid boxes below the tropopause for the mean tropospheric lifetime.

Figure 3 shows the mean tropospheric CH$_4$ lifetime of the MLO experiments, together with the fSST experiments, dependent
on the CH$_4$ scaling factor, i.e. 1 for the reference simulations, 2 for the experiments with 2$\times$CH$_4$, and 5 for those with 5$\times$CH$_4$. An almost linear relationship between the mean tropospheric CH$_4$ lifetime and the CH$_4$ scaling factor is present also in the MLO sensitivity simulations. The lifetime increase is, however, reduced by 0.30 a (increase by 2.03 a instead of 2.33 a) and 1.17 a (increase by 6.37 a instead of 7.54 a) in the MLO set-up compared to fSST when doubling and fivefolding CH$_4$,





respectively. This weaker increase is in line with a weaker decrease of tropospheric OH in the MLO sensitivity simulations
compared to fSST as obvious from Fig. 4, which shows the difference between the OH response in the MLO and in the fSST
sensitivity simulations (see Fig. 4 in Winterstein et al., 2019 and Fig. S7 for the respective response patterns of OH in the fSST
and the MLO simulations, respectively). In the troposphere the difference between the OH response in the MLO and in the fSST
experiments is hardly significant anywhere for the $2\times CH_4$ experiments, whereas it is significant in the tropics for $5\times CH_4$. The
weaker decrease of tropospheric OH in both MLO simulations is related to more strongly enhanced OH precursors ($H_2O$ and
$O_3$) in the troposphere in the MLO compared to the fSST sensitivity simulations, as will be discussed below. Additionally, the
tropospheric warming in the MLO sensitivity simulations results in a faster $CH_4$ oxidation as its reaction rate increases with
temperature. Voulgarakis et al. (2013) compared the $CH_4$ lifetime increase of two simulations, one with the full RCP8.5 climate
change signal of the year 2100 with respect to 2000, and one with $CH_4$ concentrations corresponding to 2100 RCP8.5 levels,
but climate conditions of the year 2000. They identified a weaker increase of the $CH_4$ lifetime with tropospheric warming as
well. Their difference is larger than the difference between the S2 fSST and S2 MLO lifetime responses, even though the $CH_4$
increase simulated by Voulgarakis et al. (2013) is of the same order of magnitude as in S2 fSST and S2 MLO, since the RCP8.5
scenario projects a doubling of the 2010 $CH_4$ mixing ratios at the end of the century. However, the tropospheric warming in
the RCP8.5 scenario is stronger because it includes the effects of all GHGs and not only the effect of $CH_4$. This can explain
the larger offset of the $CH_4$ lifetime response reported by Voulgarakis et al. (2013).

Please recall that we prescribe the $CH_4$ mixing ratios at the lower boundary using Newtonian relaxation. It is important to
note that the prolongation of the tropospheric $CH_4$ lifetime causes the corresponding $CH_4$ fluxes at the lower boundary to not
scale equally with the mixing ratio increase, but to increase by a smaller factor. Increasing the $CH_4$ surface mixing ratio by
a factor of 2 (5) corresponds to an increase of the $CH_4$ surface fluxes by a factor of $1.61 \pm 0.01$ ($2.91 \pm 0.01$) in the MLO
simulations, and by a factor of $1.58 \pm 0.00$ ($2.75 \pm 0.01$) in the fSST simulations (see Tab. 2). The larger increase factors
in the MLO sensitivity simulations are in line with the reduced prolongation of the tropospheric $CH_4$ lifetime compared to
the fSST experiments. The fact that the increase in emission fluxes is less than a factor of 2 or 5 suggests that enhanced $CH_4$
emissions would likewise scale the mixing ratio by a larger factor than the corresponding increase factor of the emissions. The
$CH_4$ surface fluxes that result from the nudging of the mixing ratio towards zonally averaged $CH_4$ fields are not realistic in
terms of spatial distribution, however.

**Non-linearities of $CH_4$ increase**

Winterstein et al. (2019) investigated whether the increase of atmospheric $CH_4$ follows the doubling or fivefolding for fSST
conditions linearly. Tropospheric $CH_4$ is largely controlled by the nudging at the lower boundary through mixing and responds
linearly to the increase. However, the $CH_4$ increase between 50 and 1 hPa has found to be smaller than a strictly linear relation
would predict. This indicates enhanced chemical $CH_4$ depletion in the stratosphere due to changes in the chemical composition.
Fig. 5 shows the relative difference between the annual zonal mean $CH_4$ of S2 MLO (S5 MLO) and $2\times$ ($5\times$) the zonal mean
$CH_4$ of REF MLO. The doubling or fivefolding of the reference $CH_4$ serves to emphasize regions where the increase factor of
the $CH_4$ mixing ratio deviates from 2 or 5, respectively. The response of tropospheric $CH_4$ is marginally larger than a linear





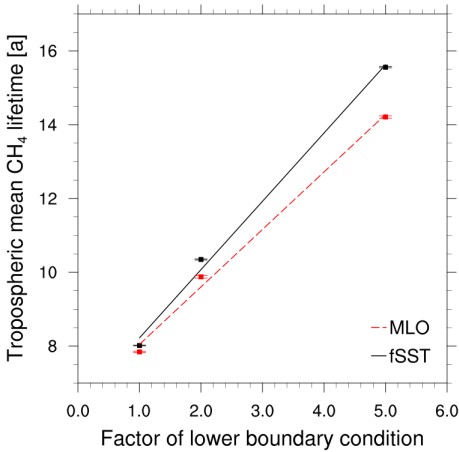

**Figure 3.** Mean tropospheric $CH_4$ lifetime with respect to the oxidation with OH versus the scaling factor of the lower boundary $CH_4$, i.e. 1 for REF, 2 for S2, 5 for S5 for the MLO (red, dashed) and the fSST (black, solid) simulations. The vertical lines indicate the 95 % confidence intervals based on annual mean values of the $CH_4$ tropospheric lifetime.

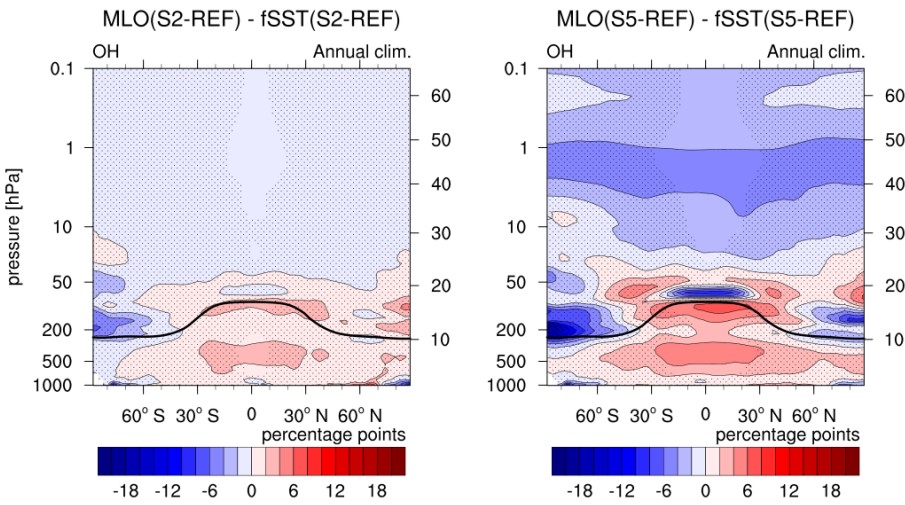

**Figure 4.** Differences between the OH response to enhanced $CH_4$ in the MLO and fSST set-ups in percentage points. To calculate the difference the relative changes of S2 fSST (left) and S5 fSST (right) are subtracted from the relative changes of S2 MLO and S5 MLO, respectively. Non-stippled areas are significant on the 95 % confidence level according to a two sided Welch's test. The solid black line indicates the climatological tropopause height of REF MLO.


**Table 2.** Increase factors of the global mean $CH_4$ surface fluxes, which correspond to increases of the $CH_4$ mixing ratios by factors of 2 or 5, respectively. The values after the $\pm$ sign are the 95 % confidence intervals of the mean calculated using Taylor expansion and assuming S2/S5 and REF fluxes to be uncorrelated as $\pm\, t_{\frac{\alpha}{2},df}\cdot\frac{\overline{x}}{\overline{y}}\cdot\sqrt{\frac{s_x^2}{N_x\cdot\overline{x}}+\frac{s_y^2}{N_y\cdot\overline{y}}}$ with the mean values of the S2/S5 and REF fluxes $\overline{x}$ and $\overline{y}$, respectively, interannual standard deviations $s_x$ and $s_y$, number of analysed years $N_x$ and $N_y$, $\alpha = 0.05$, and the degrees of freedom $df = (\frac{s_x^2}{N_x}+\frac{s_y^2}{N_y})\cdot(\frac{(\frac{s_x^2}{N_x})^2}{N_x-1}+\frac{(\frac{s_y^2}{N_y})^2}{N_y-1})^{-1}$.

|      | fSST            | MLO             |
|------|-----------------|-----------------|
| S2   | $1.58 \pm 0.00$ | $1.61 \pm 0.01$ |
| S5   | $2.75 \pm 0.01$ | $2.91 \pm 0.01$ |

increase in both MLO experiments. This is in line with the response of tropospheric $CH_4$ in the fSST simulations. As for the fSST simulations, the $CH_4$ increase in the extratropical stratosphere is weaker than a linear increase in both MLO sensitivity simulations. The non-linearity is less pronounced in the two MLO sensitivity experiments compared to the respective fSST experiments (compare with Fig. 3 in Winterstein et al., 2019) suggesting that the chemical depletion of $CH_4$ is enhanced in the MLO experiments as well, however, less strongly than in the fSST experiments.

Another aspect to note in Fig. 5 is the more than 5×$CH_4$ increase in the lowermost tropical stratosphere for S5 MLO. This feature indicates enhanced tropical upwelling, which leads to larger $CH_4$ mixing ratios in the tropical lower stratosphere. This feature is more pronounced in S5 MLO than in S5 fSST, in line with the more pronounced changes of tropical upwelling in the MLO set-up as discussed in Sect. 3.2. Furthermore, strengthening of the BDC transports $CH_4$ more efficiently to higher altitudes leading to higher $CH_4$ mixing ratios there as well. This can be one explanation for the weaker deviation from a linear $CH_4$ increase in the MLO compared to the fSST simulations. Another explanation, as already stated, is that the chemical depletion of $CH_4$ is less strongly enhanced in the MLO sensitivity simulations compared to fSST. We therefore discuss differences of the response of OH, the most important sink partner of $CH_4$, in the next paragraph.

Stratospheric OH mixing ratios increase in both simulation set-ups (fSST and MLO) at the order of 30 % for 2×$CH_4$ and 60 %–80 % for 5×$CH_4$. As shown by Winterstein et al. (2019), OH precursors ($H_2O$ and $O_3$) in the stratosphere are also affected by the $CH_4$ increase. The OH increase in the stratosphere is weaker in the MLO simulations compared to the fSST simulations (see Fig. 4). The differences are, however, small compared to the total increase of OH and mainly not significant. The difference between the two 5×$CH_4$ experiments reaches up to 5 percentage points (p.p.) in the middle stratosphere. The weaker increases of OH are presumably connected to weaker increases of SWV in the MLO simulations. The considerably weaker OH increase above the tropical tropopause in S5 MLO with respect to S5 fSST is possibly associated with a stronger $O_3$ decrease in this area in S5 MLO. Both, changes in SWV and $O_3$, will be discussed below. The weaker OH increases in the MLO sensitivity experiments with respect to fSST are in line with the smaller deviations from a linear doubling or fivefolding of the $CH_4$ mixing ratio in the stratosphere (see Fig. 5). We conclude that the strengthening of the $CH_4$ oxidation resulting from increases of the OH mixing ratio is weaker in the MLO experiments, but still present.



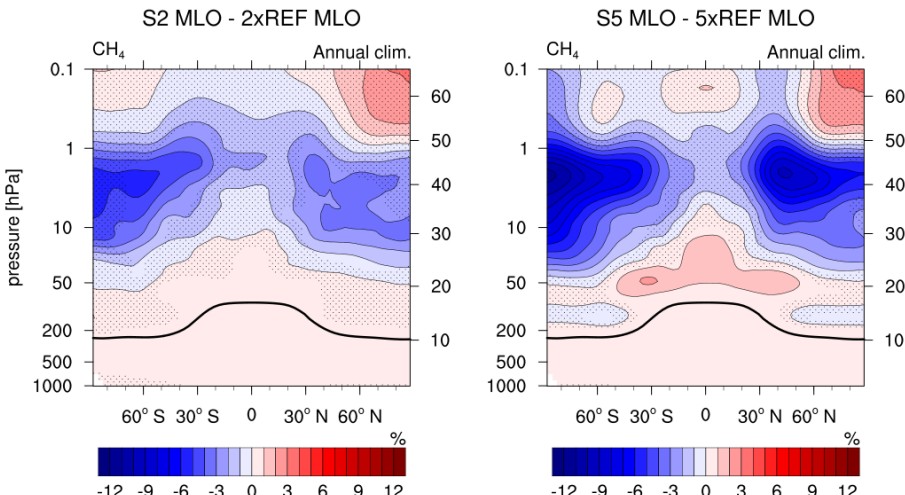

**Figure 5.** Relative differences between the annual zonal mean $CH_4$ of the sensitivity simulations S2 MLO and $2\times$ REF MLO (left) and S5 MLO and $5\times$ REF MLO (right) in %. Non-stippled areas are significant on the 95 % confidence level according to a two sided Welch's test. The solid black line indicates the climatological tropopause height of REF MLO.

**Water vapour**

$H_2O$ is a precursor of OH and its abundance is also influenced by $CH_4$ mixing ratios. Winterstein et al. (2019) reported a steady increase of $H_2O$ with height for the $CH_4$ doubling and fivefolding experiments with prescribed SSTs and SICs. Figure 6

shows the difference of the $H_2O$ response between the MLO and the fSST simulations (see Fig. 5 in Winterstein et al., 2019 and Fig. S8 for the respective response patterns of $H_2O$ in the fSST and the MLO simulations, respectively). As the saturation vapour pressure increases with temperature, the warming of the troposphere in the MLO simulations consistently leads to a stronger increase of the tropospheric $H_2O$ mixing ratio in comparison with the respective fSST simulation. The maximum difference between MLO and fSST can be found in the upper tropical troposphere and extratropical lowermost stratosphere

and reaches 11 p.p. (35 p.p.) for the $2\times$ ($5\times$) $CH_4$ experiments.

In the middle and upper stratosphere, the $H_2O$ increase is about 5 p.p. (15 p.p.) weaker in the S2 MLO (S5 MLO) sensitivity simulation compared to S2 fSST (S5 fSST). This reduction is significant, but small compared to the relative increase of SWV of around 50 % for both $2\times CH_4$, and 250 % for both $5\times CH_4$ experiments. The amount of tropospheric $H_2O$ transported into the stratosphere is largely determined by the cold point temperature (CPT) (e.g., Randel and Park, 2019). Furthermore, the

oxidation of $CH_4$ is an important in-situ source of SWV (Hein et al., 2001; Rohs et al., 2006; Frank et al., 2018). The SWV mixing ratio at a given location and time can be approximated as the sum of these two terms (Austin et al., 2007; Revell et al., 2016). We calculate the amount of tropospheric $H_2O$ entering the stratosphere as the tropical (10°N–10°S) mean $H_2O$ mixing ratio at 70 hPa following Revell et al. (2016). The $H_2O$ entry mixing ratio increases by about 10 % (40 %) in the $CH_4$ doubling (fivefolding) experiments (both MLO and fSST). The relative increases are insignificantly higher in both MLO experiments





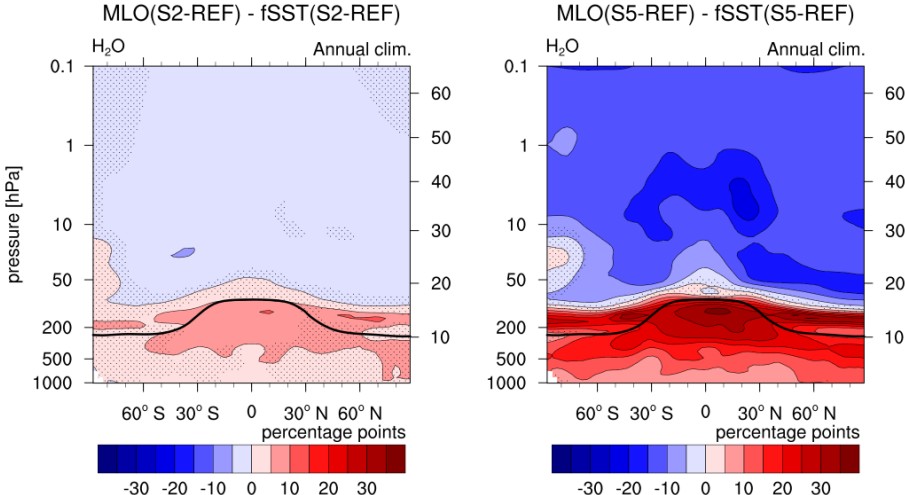

**Figure 6.** Differences between the $H_2O$ response to enhanced $CH_4$ in the MLO and fSST set-ups in percentage points. To calculate the difference the relative changes of S2 fSST (left) and S5 fSST (right) are subtracted from the relative changes of S2 MLO and S5 MLO, respectively. Non-stippled areas are significant on the 95 % confidence level according to a two sided Welch's test. The solid black line indicates the climatological tropopause height of REF MLO.

compared to the respective fSST experiment. Furthermore, the zonal mean tropical CPT increases in all sensitivity simulations (see Fig. S9). The magnitude and the latitude dependence of the CPT changes are very similar for both doubling and both fivefolding experiments, although slightly larger for the MLO experiments in line with the changes of the $H_2O$ entry mixing ratio. Changes of the amount of tropospheric $H_2O$ entering the stratosphere can therefore not explain the differences in the SWV response between MLO and fSST in the middle and upper stratosphere. The increases of the $H_2O$ entry mixing ratio

and the CPT are both slightly stronger in the MLO experiments and would therefore suggest a stronger increase of SWV in the MLO experiments. On the contrary, the increases of SWV are weaker in the middle and upper stratosphere in the MLO experiments compared to fSST. The contribution of the $CH_4$ oxidation on SWV can explain these weaker increases of SWV in the MLO experiments. The strengthening of the $CH_4$ oxidation in the stratosphere is weaker in the MLO experiments resulting likewise in a weaker increase of SWV produced by $CH_4$ oxidation.

What remains to be explained is the reason for the weaker strengthening of the $CH_4$ oxidation in the MLO setup compared to fSST. Strengthened tropical upwelling as shown in Sect. 3.2 transports $CH_4$ into the stratosphere more efficiently and would be expected to lead to higher rates of the $CH_4$ oxidation (Austin et al., 2007). However, as the strengthening of the $CH_4$ oxidation is weaker in the MLO experiments, $CH_4$ itself seems not to be the limiting factor here. The abundance of SWV feeds back on OH and therefore also on the efficiency of the $CH_4$ oxidation. However, the increase of SWV seems to be rather a result of the

strengthened $CH_4$ oxidation here, as the increase of $H_2O$ entering the stratosphere is higher in the MLO experiments compared to fSST.





**Ozone**

The other important precursor of OH is $O_3$, the abundance of which is also influenced by $CH_4$. The stratospheric $O_3$ response pattern in the MLO experiments, namely $O_3$ reduction in the lowermost tropical stratosphere, $O_3$ increase up to approximately

2 hPa, and $O_3$ decrease above, is qualitatively consistent with the fSST simulations (compare Fig. 7 in Winterstein et al., 2019 and Fig. S10). Winterstein et al. (2019) gave a detailed explanation of the processes leading to the resulting $O_3$ pattern that is also valid for the MLO simulations. When subtracting the fSST response from the MLO response, the extra effect of tropospheric warming becomes apparent. The resulting patterns for S2 and S5 are shown in Fig. 7. A dominant feature is the stronger decrease of $O_3$ in the lowermost tropical stratosphere in S5 MLO compared to S5 fSST of up to 18 p.p.. This difference

also exists between the S2 simulations, albeit weaker (4 p.p.). The more strongly decreasing $O_3$ mixing ratios in MLO indicate that the transport of $O_3$ poor air from the troposphere into the stratosphere is intensified in the MLO simulations. The increases of $O_3$ in the southern polar middle stratosphere in S2 MLO, and in both polar regions in S5 MLO are more pronounced with respect to the respective fSST experiment. This indicates more strongly enhanced meridional transport in the MLO experiments. Both patterns are in line with the strengthening of the residual mean circulation as discussed in Sect. 3.2.

In the tropospheric $O_3$ response pattern (shown in Fig. S10) any $O_3$ feedback from tropospheric warming is superimposed by chemical influences of $CH_4$. Therefore, the pattern is fundamentally different from $O_3$ changes in global warming simulations driven by $CO_2$ increases (see Fig. 1 (a) in Dietmüller et al., 2014, Fig. 3 (a) in Nowack et al., 2018, and Fig. 1 (a) - (c) in Chiodo and Polvani, 2019), where direct chemical impacts are weak. However, if the $O_3$ response to slow climate feedbacks induced by enhanced $CH_4$ is separated from rapid adjustments (Fig. 7), a similar pattern to the $O_3$ response induced by enhanced

$CO_2$ arises. An exception is the increase of $O_3$ above 30 hPa that results from a slower chemical depletion of $O_3$ caused by stratospheric radiative cooling (Dietmüller et al., 2014), which develops on the timescale of rapid radiative adjustments. A deceleration of the chemical $O_3$ destruction in the middle stratosphere is also present in the $CH_4$ driven experiments resulting mainly from radiative cooling induced by adjustments of SWV and $O_3$ (see Fig. 8 (e) and (f) in Winterstein et al., 2019), but cancels out in Fig. 7.

### 3.3.2   Radiative impact, surface temperature response and climate sensitivity

In Winterstein et al. (2019) the total RI has been separated into the individual contributions of the species $CH_4$, SWV, and $O_3$, an analysis we extend hereafter to the MLO simulations. Note, that we adopt the definition of Winterstein et al. (2019) concerning the RI, which indicates the radiative flux imbalance between the sensitivity and the reference simulation.

In Table 3 we summarize the RI of the most important species in both the fSST and the MLO simulations. The individ-

ual contributions to the RI have been calculated with the submodel RAD (Dietmüller et al., 2016) in separate simulations (S2 fSST*, S5 fSST*, S2 MLO* and S5 MLO*; see Sect. 2). We further separate the $H_2O$ and $O_3$ contribution into tropospheric and stratospheric RI, respectively. The RIs of $CH_4$ and $O_3$ show only small differences between fSST and MLO. This implies that SST-driven climate feedbacks on these constituents do not substantially alter their RI contribution in our simulation set-up. As expected, the RI of tropospheric $H_2O$ increases substantially (from $0.08 \pm 0.05$ Watt per square meter (W m$^{-2}$)



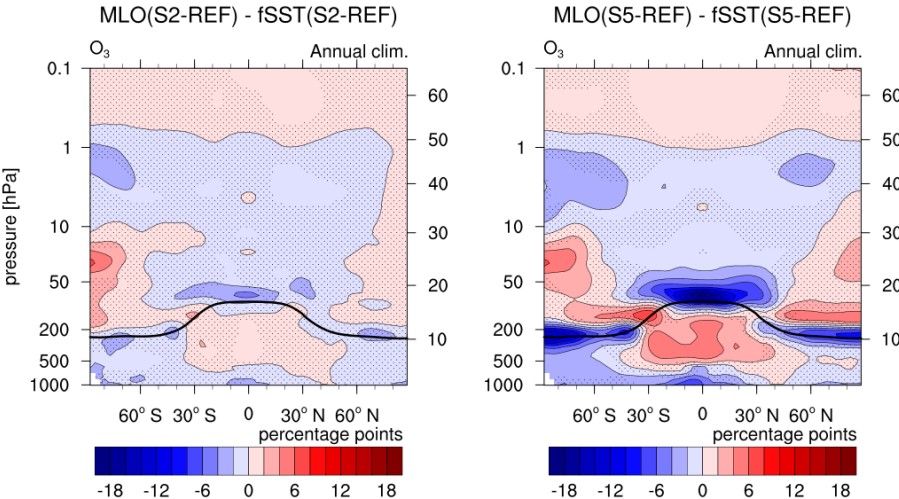

**Figure 7.** Differences between the $O_3$ response to enhanced $CH_4$ in the MLO and fSST set-ups in percentage points. To calculate the difference the relative changes of S2 fSST (left) and S5 fSST (right) are subtracted from the relative changes of S2 MLO and S5 MLO, respectively. Non-stippled areas are significant on the 95 % confidence level according to a two sided Welch's test. The solid black line indicates the climatological tropopause height of the REF MLO.

to $0.72 \pm 0.04$ W m$^{-2}$ for 2×$CH_4$ and from $0.30 \pm 0.06$ W m$^{-2}$ to $2.23 \pm 0.06$ W m$^{-2}$ for 5×$CH_4$). The RI of stratospheric $H_2O$ increases as well, which is mostly influenced by the increase in SWV in the lowermost stratosphere due to transport of moist air from the tropical troposphere into the stratosphere (see Fig. 6).

The global mean surface temperature responses in the MLO experiments for 2× and 5×$CH_4$ are $0.42 \pm 0.05$ K and $1.28 \pm 0.04$ K, respectively. Under the reasonable assumption that the total RIs from the fSST experiments represent the corresponding
ERFs with chemical rapid adjustments included (Winterstein et al., 2019), we calculate the climate sensitivity parameters $\lambda$ as $0.61 \pm 0.17$ K W$^{-1}$ m$^2$ and $0.72 \pm 0.07$ K W$^{-1}$ m$^2$, respectively. The estimate of $\lambda$ corresponding to 5×$CH_4$ compares well with the climate sensitivity parameter $\lambda_{adj}$ of $0.73$ K W$^{-1}$ m$^2$ from Rieger et al. (2017) corresponding to a 1.2×$CO_2$ experiment with EMAC with a RF of $1.06$ W m$^{-2}$, which is comparable to the RIs in the present experiments. The agreement of the climate sensitivity parameters for $CH_4$- and $CO_2$-forcing suggests an efficacy of $CH_4$ ERF close to one. The estimate of
$\lambda$ for 2×$CH_4$ is smaller than the value from Rieger et al. (2017), but the difference is insignificant as a consequence of large statistical uncertainty.

In a recent multimodel comparison, the multimodel mean efficacy of $CH_4$ was found to be smaller than one, however, with a large intermodel spread ranging from 0.56 to 1.15 (Richardson et al., 2019). Modak et al. (2018) found a $CH_4$ efficacy of 0.81 for $CH_4$ for a simulation with a $CH_4$ increase comparable to S5. They identified $CH_4$ shortwave (SW) absorption and related
warming of the lower stratosphere and upper troposphere as reason for the smaller efficacy of $CH_4$. Our simulation set-up does not account for SW absorption of $CH_4$. The climate sensitivity and efficacy estimates of Modak et al. (2018) and Richardson





et al. (2019) do not include chemical feedbacks of $O_3$ and SWV induced by $CH_4$. They also do not provide a robust indication that the $CH_4$ efficacy is significantly larger or smaller than unity in their framework, as the inter-model spread reported by (Richardson et al., 2019) is so large. Estimating a reasonable climate sensitivity value from our simulations in an interactive

chemistry framework, requires that rapid adjustments from SWV and $O_3$ are included in the effective $CH_4$ forcing. If this is done, these simulations do not point at a significant climate sensitivity deviation from the $CO_2$ behavior either.

**Table 3.** An estimation of individual RI contributions in [W m$^{-2}$] of the changes in the chemical species $CH_4$, $H_2O$ and $O_3$. Values are calculated using the RAD submodel (Dietmüller et al., 2016) in separate simulations (S2 fSST*, S5 fSST*, S2 MLO* and S5 MLO*, see Sect. 2) using 20 years climatologies of the individual species from the corresponding reference and sensitivity simulation experiments fSST and MLO. The lower part shows the global mean 2 m air temperature changes of S2 MLO and S5 MLO with respect to REF MLO and the total RIs of S2 fSST and S5 fSST. From these temperature changes and total RIs the climate sensitivity parameter $\lambda$ is calculated as $\lambda = \Delta T_{MLO}$ / total RI$_{fSST}$.

| Simulation | CH$_4$ | trop. H$_2$O | strat. H$_2$O | total H$_2$O | trop. O$_3$ | strat. O$_3$ | total O$_3$ |
|---|---|---|---|---|---|---|---|
| S2 fSST* | $0.23 \pm 0.01$ | $0.08 \pm 0.05$ | $0.15 \pm 0.00$ | $0.24 \pm 0.05$ | $0.22 \pm 0.01$ | $0.06 \pm 0.01$ | $0.27 \pm 0.02$ |
| S5 fSST* | $0.51 \pm 0.02$ | $0.30 \pm 0.06$ | $0.55 \pm 0.01$ | $0.85 \pm 0.06$ | $0.56 \pm 0.02$ | $0.20 \pm 0.02$ | $0.76 \pm 0.02$ |
| S2 MLO* | $0.23 \pm 0.01$ | $0.72 \pm 0.04$ | $0.19 \pm 0.00$ | $0.91 \pm 0.04$ | $0.22 \pm 0.01$ | $0.06 \pm 0.00$ | $0.28 \pm 0.01$ |
| S5 MLO* | $0.52 \pm 0.02$ | $2.23 \pm 0.06$ | $0.65 \pm 0.01$ | $2.87 \pm 0.07$ | $0.57 \pm 0.02$ | $0.19 \pm 0.01$ | $0.76 \pm 0.02$ |

| | $\Delta T_{MLO}$ [K] | total RI$_{fSST}$ [W m$^{-2}$] | $\lambda$ [K W$^{-1}$ m$^2$] |
|---|---|---|---|
| S2 | $0.42 \pm 0.05$ | $0.69 \pm 0.16$ | $0.61 \pm 0.17$ |
| S5 | $1.28 \pm 0.04$ | $1.79 \pm 0.17$ | $0.72 \pm 0.07$ |

The values after the $\pm$ sign are the 95 % confidence intervals of the mean.

For $\lambda$ the confidence intervals are calculated using Taylor expansion and assuming $\Delta T_{MLO}$ and total RI$_{fSST}$ to be uncorrelated as $\pm t_{\frac{\alpha}{2},df} \cdot \frac{\overline{x}}{\overline{y}} \cdot \sqrt{\frac{s_x^2}{N_x \cdot \overline{x}} + \frac{s_y^2}{N_y \cdot \overline{y}}}$ with the mean values of $\Delta T_{MLO}$ and total RI$_{fSST}$ $\overline{x}$ and $\overline{y}$, respectively, interannual standard deviations $s_x$ and $s_y$, number of analysed years $N_x$ and $N_y$, $\alpha = 0.05$, and the degrees of freedom $df = (\frac{s_x^2}{N_x} + \frac{s_y^2}{N_y}) \cdot (\frac{(\frac{s_x^2}{N_x})^2}{N_x - 1} + \frac{(\frac{s_y^2}{N_y})^2}{N_y - 1})^{-1}$.

### 3.3.3 Radiatively and dynamically driven atmospheric temperature response

Fig. 8 shows the differences of temperature response between the MLO and the fSST simulations. As expected, tropospheric warming is significantly stronger in the MLO experiments, since the tropospheric temperature change is largely suppressed

in the simulations with prescribed SSTs and SICs. In the stratosphere, radiatively and dynamically driven effects contribute to differences in the temperature change patterns between MLO and fSST, as will be shown in the following. Note again that changes in the chemical composition resulting from a change in circulation (i.e. transport) are included in the radiatively driven effects by our definition.

     Following Winterstein et al. (2019) we calculate the stratospheric adjusted temperature response to changes in $CH_4$, tropo-

spheric and stratospheric $H_2O$, and tropospheric and stratospheric $O_3$, as well as their individual contributions for S2 MLO and





S5 MLO (see Fig. S11 for simulation S2 MLO and Fig. 9 for simulation S5 MLO). The difference of adjusted stratospheric temperature response between S5 MLO and S5 fSST is shown in Fig. 10 (for S2 see Fig. S12). This difference is small for $CH_4$ and tropospheric $O_3$ (see Fig. 10 (b) and (g)). Figure 10 (d) confirms the stratospheric radiative cooling effect of increased humidity in the troposphere in S5 MLO, although the effect is quantitatively small. The adjusted stratospheric temperature
response pattern induced by SWV in S5 MLO is similar to S5 fSST. However, the stronger increases of SWV in S5 MLO result in more pronounced cooling in the lowermost stratosphere, whereas the reduced increases above consistently result in reduced cooling (see Fig. 10 (e)). The stronger decrease of $O_3$ in the tropical lower stratosphere in S5 MLO (see Fig. 7) leads to stronger cooling in this region as shown in Fig. 10 (h). These results also apply qualitatively to the comparison of S2 MLO and S2 fSST (see Fig. S12), but the magnitude of the differences is smaller. The effects from SWV and stratospheric $O_3$ dominate
the differences of stratospheric adjusted temperature between S5 MLO and S5 fSST (compare Fig. 10 (a)). In addition, the resulting more pronounced cooling in the lowermost stratosphere in the MLO simulations is apparent in the difference between the overall temperature responses of MLO and fSST in Fig. 8.

By calculating the difference between the total temperature response in the regular simulations and the sum of the individual contributions of $CH_4$, $H_2O$ and $O_3$ to the adjusted stratospheric temperatures, we attempt to identify the dynamical effect
$(\Delta\tilde{T}_{\mathrm{dyn.}})$ in the stratospheric temperature response as

$$\Delta\tilde{T}_{\mathrm{dyn.}} = \Delta T(\mathrm{SX\text{-}REF}) - \Delta T_{\mathrm{addst}}(\mathrm{SX^*\text{-}REF^*}) \qquad (2)$$

with X being either 2 or 5. A similar approach was, for example, used by Rosier and Shine (2000) and Schnadt et al. (2002) to distinguish between the radiative impact of trace gases and dynamical contributions to the total temperature response.

Fig. 11 shows the annual mean of $\Delta\tilde{T}_{\mathrm{dyn.}}$ for all four sensitivity simulations. It is mostly not significant for S2 fSST and
S5 fSST in the stratosphere suggesting that dynamical effects play a minor role in the temperature response in these simulations as already indicated by Winterstein et al. (2019). However, immediately above the tropical tropopause centered at the equator $\Delta\tilde{T}_{\mathrm{dyn.}}$ indicates warming for both, S2 fSST and S5 fSST. In austral winter (JJA), $\Delta\tilde{T}_{\mathrm{dyn.}}$ shows significant cooling in the southern polar stratosphere for S2 fSST and S5 fSST. The cooling extends into austral spring (SON), but gradually weakens as time proceeds (see Fig. S15 and Fig. S16). These temperature changes can be associated to the strengthening of the SH
stratospheric winter polar vortex (see Fig. S18), which leads to enhanced isolation of airmasses and stronger cooling. The stratospheric polar vortex in boreal winter DJF accelerates in both fSST sensitivity simulations as well (see Fig. S17).

The pattern of $\Delta\tilde{T}_{\mathrm{dyn.}}$ for S5 MLO (Fig. 11 (d)) displays a near-symmetrical behavior around the equator. It comprises of two warming patches in the lower stratosphere - unlike S5 fSST not centered at the equator, but at around 30°S or 30°N -, as well as cooling in the tropics and warming in the extratropics in the middle stratosphere. The warming patches in the lower
stratosphere are present in all seasons, whereas the pattern of cooling in the tropics and warming in the extratropics above is shifted to the respective winter hemisphere (compare Fig. S13 and Fig. S15). For S2 MLO, the warming patches in the lower stratosphere are also present in the pattern of $\Delta\tilde{T}_{\mathrm{dyn.}}$. Apart from that, the annual mean $\Delta\tilde{T}_{\mathrm{dyn.}}$ is mostly not significant for S2 MLO. However, the pattern of cooling in the tropics and warming in the extratropics is indicated in boreal autumn (SON) and winter (DJF) for S2 MLO as well.





We associate the main component of the $\Delta\tilde{T}_{\mathrm{dyn.}}$ pattern of the MLO experiments with the strengthening of the BDC as discussed in Sect. 3.2. Strengthened downwelling in the subtropical and extratropical lower stratosphere results in adiabatic warming in this region in both hemispheres throughout the year. These temperature changes can therefore be associated with the intensification of the shallow branch of the BDC (Plumb, 2002; Birner and Bönisch, 2011). The patterns are present in S2 MLO and S5 MLO. Adiabatic cooling in the tropical middle and upper stratosphere, as well as a respective adiabatic

warming in the extratropical and polar winter stratosphere indicate the strengthening of the deep branch of the BDC, more pronounced in S5 MLO than in S2 MLO. The strengthening of the BDC would be expected to result in adiabatic cooling directly above the tropopause from increased tropical upwelling. This effect seems to be masked by other processes in Fig. 11. These could be advection or mixing of warm air from the troposphere, or increased longwave (LW) radiation from the warmer troposphere and potentially more LW absorption in the lowest stratosphere. Lin et al. (2017) found the latter effect to cause

strong warming in the tropical tropopause layer. This radiative effect is not accounted for in $\Delta T_{\mathrm{addst}}$(SX*-REF*), which is the sum of the individual contributions of radiatively active gases to the adjusted stratospheric temperatures. Furthermore, mixing with air out of the upper tropical troposphere could also contribute to the warming patches in the subtropical and extratropical lower stratosphere. This region is particularly affected by mixing (Dietmüller et al., 2018; Eichinger et al., 2019) and mixing itself can also be influenced by climate change (Eichinger et al., 2019).

The deep branch of the residual mean circulation is closely linked to the strength of the winter stratospheric polar vortex. An increase in the poleward flow and in downwelling at higher latitudes is accompanied with a slow down of the stratospheric polar vortex (Kidston et al., 2015, and references therein). The S5 MLO response of zonal mean winds shows indeed an easterly change of the stratospheric polar vortex in boreal winter (DJF) (see Fig. S17). The respective response for S2 MLO is not significant, but decelerating, too. The SH stratospheric polar vortex strengthens for S2 MLO, but less than in S2 fSST.

Nevertheless, the response of stratospheric zonal winds in both MLO experiments is substantially different from fSST in the SH as well.

    The easterly change of polar stratospheric zonal winds in the NH during DJF is consistent with the response of the stratospheric polar vortex in CMIP5 global warming simulations (Manzini et al., 2014; Karpechko and Manzini, 2017). Moreover, differences between the fSST and MLO response signals of stratospheric zonal winds during DJF are qualitatively consistent

with the results of Karpechko and Manzini (2017). They identified, on the one hand, a deceleration of the stratospheric polar vortex and associated warming in the polar stratosphere in simulations driven by higher SSTs (comparable to the MLO experiments), and, on the other hand, a strengthened and cooled stratospheric polar vortex in simulations driven by $CO_2$ increase and suppressed tropospheric warming (comparable to the fSST experiments). Karpechko and Manzini (2017) suggested that tropospheric warming and associated strengthening of subtropical winds lead to enhanced wave activity. In S5 MLO subtropical

winds strengthen indicating that similar processes might act in our simulations. However, a detailed analysis of wave activity is beyond the scope of this study.

    In summary, SST-driven climate feedbacks affect the chemical composition. The differences in stratospheric temperature adjustment between MLO and fSST (see Fig. 10) reflect radiative impacts of these composition changes on stratospheric temperature. Additionally, the patterns of $\Delta\tilde{T}_{\mathrm{dyn.}}$ suggest that dynamical effects have changed significantly in the MLO simulations





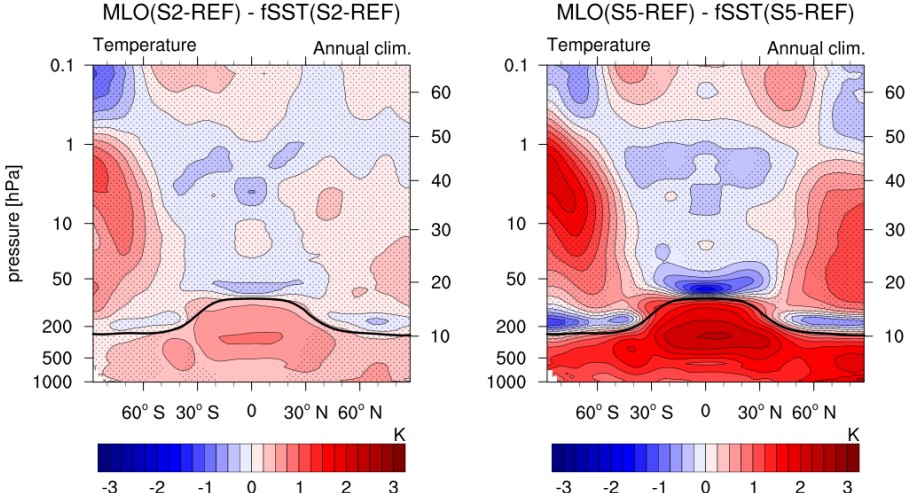

**Figure 8.** Differences between the temperature response to enhanced $CH_4$ in the MLO and fSST set-ups in K. To calculate the difference the absolute changes of S2 fSST (left) and S5 fSST (right) are subtracted from the absolute changes of S2 MLO and S5 MLO, respectively. Non-stippled areas are significant on the 95 % confidence level according to a two sided Welch's test. The solid black line indicates the climatological tropopause height of the REF MLO.

with respect to fSST. The dynamical temperature response effect for S5 MLO is consistent with the strengthening of the BDC. Dynamic heating counteracts the radiative cooling in the extratropical middle and upper stratosphere and in the subtropical lower stratosphere in S5 MLO. This results in reduced cooling in these regions in S5 MLO in Fig. 8, which is not significant on annual average, but in the respective winter hemispheres (not shown). $\Delta \tilde{T}_{\mathrm{dyn.}}$ for S2 MLO indicates strengthening of mainly the shallow branch of the BDC.



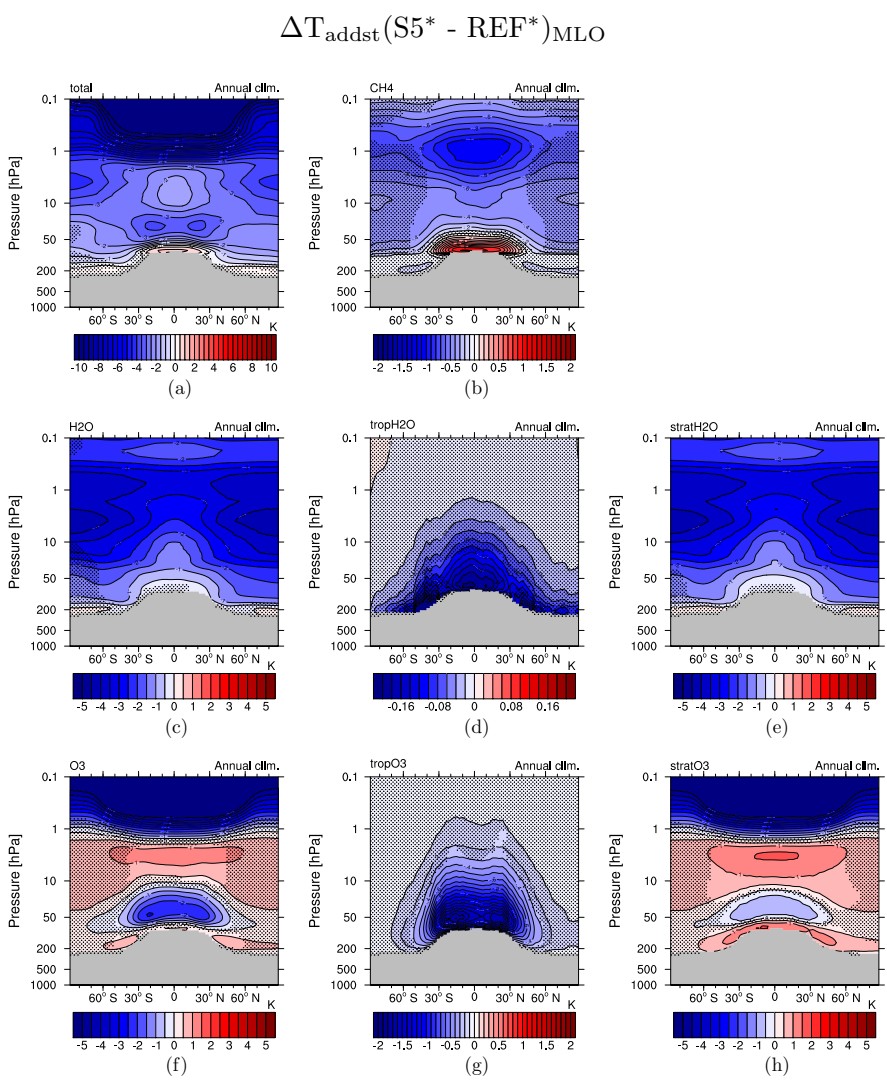

**Figure 9.** Stratospheric temperature adjustment radiatively induced by individual species changes in simulation S5 MLO ($5 \times CH_4$): (a) $CH_4$, $H_2O$ and $O_3$ combined, (b) $CH_4$, (c) $H_2O$, (d) tropospheric $H_2O$ only, (e) stratospheric $H_2O$ only (SWV), (f) $O_3$, (g) tropospheric $O_3$ only and (h) stratospheric $O_3$ only. Note the different colour bars in panels (a), (b), (d) and (g).



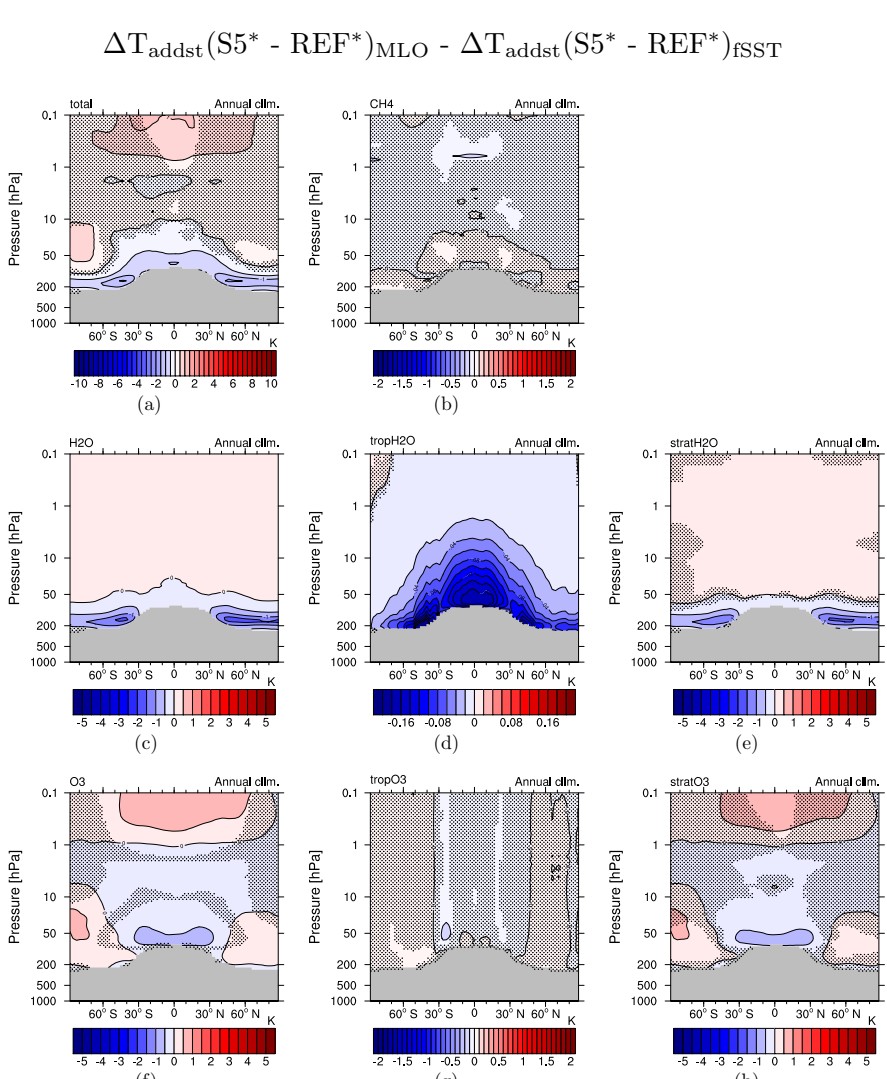

**Figure 10.** Difference between stratospheric temperature adjustment in simulations S5 MLO and S5 fSST (5×CH$_4$) radiatively induced by individual species changes: (a) CH$_4$, H$_2$O and O$_3$ combined, (b) CH$_4$, (c) H$_2$O, (d) tropospheric H$_2$O only, (e) stratospheric H$_2$O only (SWV), (f) O$_3$, (g) tropospheric O$_3$ only and (h) stratospheric O$_3$ only. Note the different colour bars in panels (a), (b), (d) and (g).



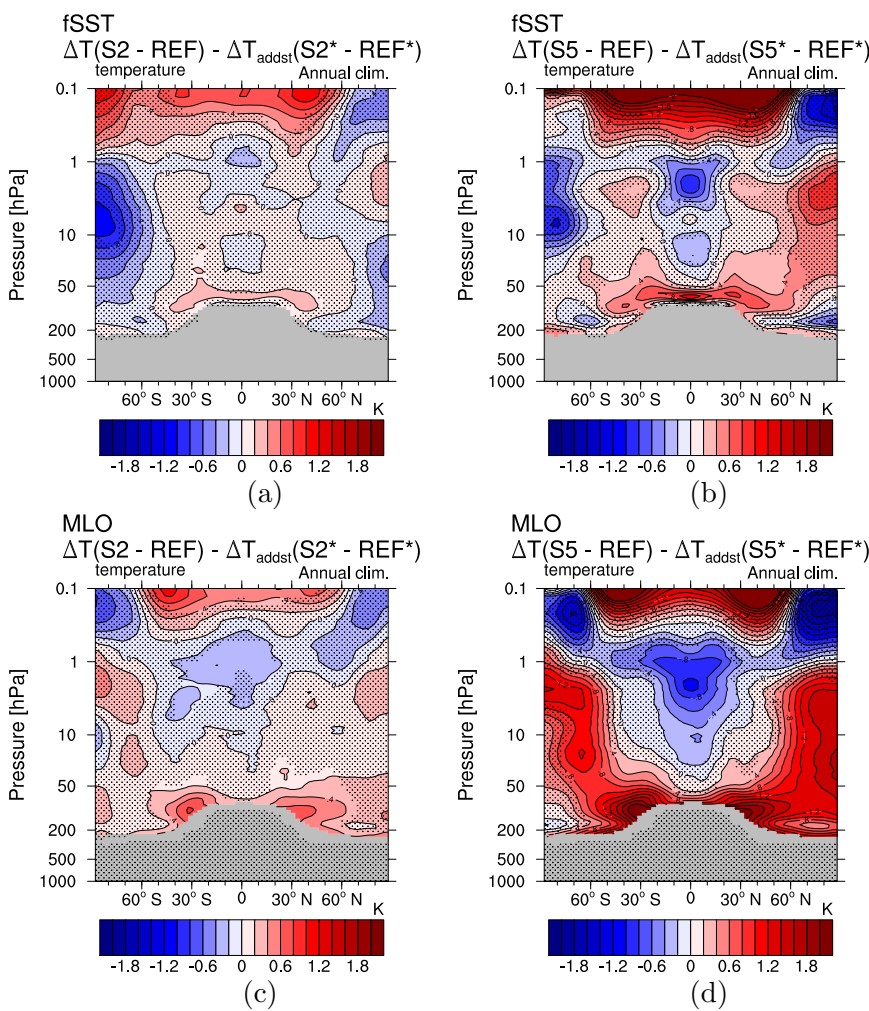

**Figure 11.** Dynamical temperature response effect of the simulations (a) S2 fSST, (b) S5 fSST, (c) S2 MLO, (d) S5 MLO. The dynamical effect is calculated as the difference between the temperature response in the regular simulations ($\Delta$T(SX-REF) with X either 2 or 5) and the sum of the individual contributions of $CH_4$, $H_2O$ and $O_3$ to the adjusted stratospheric temperatures ($\Delta T_{addst}$(SX*-REF*)) with X either 2 or 5).



## 4 Summary and Conclusions

While it has been long-since acknowledged that the net RF of $CH_4$ includes substantial contributions from $O_3$ and SWV (e.g., IPCC, 2013, Fig. 8.17), it is still common to consider climate feedbacks and climate sensitivity of $CH_4$ in comparison to $CO_2$ without accounting for these additional radiative components (Modak et al., 2018; Smith et al., 2018; Richardson et al., 2019). Our study provides a quantification of SST-driven slow radiative feedbacks from $CH_4$, $O_3$ and associated SWV changes in climate sensitivity simulations forced by twofold or fivefold $CH_4$ increase, extending the work of Winterstein et al. (2019) on the respective rapid radiative adjustments.

The strongly enhanced $CH_4$ mixing ratios cause enhanced depletion of OH in the troposphere. Tropospheric warming, in contrast, results in enhanced OH precursors and causes the reduction of OH in the troposphere to be weaker than in the prescribed SST simulations analysed by Winterstein et al. (2019). Additionally, the acceleration of the $CH_4$ oxidation at higher temperatures leads to a more efficient depletion of $CH_4$ in a warming troposphere. This so called climate offset results in a reduced prolongation of the tropospheric $CH_4$ lifetime and is consistent with previous CCM studies (Voulgarakis et al., 2013). The prolonged tropospheric $CH_4$ lifetime has the effect that the corresponding $CH_4$ surface fluxes increase by a smaller factor than the mixing ratio.

Changes in the stratospheric circulation can be clearly identified in the sensitivity simulations that include SST-driven climate feedbacks, on top of the quasi-instantaneous response analysed by Winterstein et al. (2019). Tropospheric warming leads to the acceleration of the BDC in our sensitivity simulations as expected from climate change scenario calculations (Butchart, 2014). In the lower tropical stratosphere, both the decrease of $O_3$ and the associated cooling, and the increase in $CH_4$ become more distinct, which reflects the more pronounced acceleration of tropical upwelling induced by a warming troposphere. The strengthening of the BDC also manifests in the temperature response. Whereas the stratospheric polar vortices in both winter hemispheres strengthen in the experiments with prescribed SSTs and SICs, polar stratospheric zonal winds decelerate in northern winter in the sensitivity simulation that include tropospheric warming consistent with the response in CMIP5 global warming simulations (Manzini et al., 2014; Karpechko and Manzini, 2017).

As a result of tropical upper troposphere moistening, increased tropical upwelling and more pronounced warming of the cold point, the transport of tropospheric $H_2O$ into the lower stratosphere is more strongly enhanced in the sensitivity simulations that include SST-driven climate feedbacks, resulting in a stronger increase of SWV in the lower extratropical stratosphere. In the middle and upper stratosphere, where $CH_4$ oxidation makes an important contribution to SWV, the increase of SWV is weakened in the present sensitivity simulations compared to the quasi-instantaneous response. Less pronounced increases of stratospheric OH in response of the slow adjustments in comparison to the quasi-instantaneous response cause the depletion of $CH_4$ to be weaker, and thus the in situ source of SWV as well.

The contribution of SST-driven climate feedbacks to the total $CH_4$ induced $O_3$ response shows remarkable similarities to the $O_3$ response to climate feedbacks in $CO_2$-forced climate change simulations (Dietmüller et al., 2014; Nowack et al., 2018; Chiodo and Polvani, 2019). The consistency between the $O_3$ feedbacks resulting from these different forcing agents encourages the separation of the $O_3$ response patterns into rapid adjustments and climate feedbacks in future studies. Rapid adjustments





are specific to the forcing, whereas climate feedbacks are driven by surface temperature changes and are therefore expected to
be less dependent on the forcing agent (Sherwood et al., 2015).

The doubled and fivefold $CH_4$ mixing ratios result in global mean surface temperature changes of $0.42 \pm 0.05$ K and
$1.28 \pm 0.04$ K, respectively. We estimate the corresponding climate sensitivity parameters $\lambda$ using these temperature changes
and the respective RIs from $CH_4$ and the respective chemical adjustments, as determined by Winterstein et al. (2019), that can
well be interpreted as the corresponding ERFs. The respective estimate of $\lambda$ for $5 \times CH_4$ compares well with an estimate from
$CO_2$-driven climate change simulations with EMAC with comparable magnitude of RI (Rieger et al., 2017), suggesting an
efficacy of $CH_4$ ERF close to one. The estimate of $\lambda$ corresponding to $2 \times CH_4$ is smaller than the respective value for $5 \times CH_4$,
but has a large uncertainty. Considering the large uncertainty and intermodel spread (Richardson et al., 2019) of this parameter,
we conclude that a more targeted experimental design is necessary to exactly quantify the effect of chemical feedbacks on the
climate sensitivity in $CH_4$-driven scenarios and its efficacy with respect to $CO_2$ forcing.

The RIs from the purely SST-driven response of $CH_4$ and $O_3$ are small. The RIs resulting from changes of tropospheric and
stratospheric $H_2O$ are enlarged by SST-driven climate feedbacks. Increased tropospheric humidity in a warming troposphere
enhances the RI. The reason for the enlarged RI from SWV is its more pronounced increase in the lower stratosphere, where
its changes dominate the induced RI (Solomon et al., 2010). As the increase of SWV in this region is likely induced by
transport from the warmer tropical troposphere, this part of the RI increase cannot be regarded to be a chemically induced
rapid adjustment. The associated responses of stratospheric adjusted temperatures from the purely SST-driven response are
dominated by the just explained changes of SWV and by decreases of stratospheric $O_3$ in the lowermost tropical stratosphere.
It is worth noting, that tropospheric $CH_4$ mixing ratios do not respond to changes in tropospheric sinks (e.g. OH) in the used
simulation set-up, as its mixing ratio is prescribed at the lower boundary. The prolongation of the tropospheric $CH_4$ lifetime
indicates a positive feedback on the $CH_4$ mixing ratio, and thus on the induced RI. In a future study, climate change scenario
simulations conducted with a CCM with realistic $CH_4$ emission fluxes are planned to quantify this chemical feedback of $CH_4$.

In the present study we are able for the first time to quantify the effects of slow climate feedbacks on the chemical composition and circulation in $CH_4$-forced climate change scenarios and further evaluate them in comparison to the quasi-instantaneous atmospheric response.

*Code and data availability.* The Modular Earth Submodel System (MESSy) is continuously developed and applied by a consortium of
institutions. The usage of MESSy and access to the source code is licensed to all affiliates of institutions, which are members of the MESSy
Consortium. Institutions can become members of the MESSy Consortium by signing the MESSy Memorandum of Understanding. More
information can be found on the MESSy Consortium website (https://www.messy-interface.org/, last access: 27 May 2020, Jöckel P. and
the MESSy Consortium). Furthermore the exact code version used to produce the simulation results is archived at the German Climate
Computing Center (DKRZ) and can be made available to members of the MESSy community upon request. The simulation results are also
archived at DKRZ and are available opon request.



## Appendix A

The MLO simulations were carried out with a more recent MESSy version with regard to the fSST simulations (2.54.0 instead of 2.52). This involves changes to the chemistry module MECCA (Sander et al., 2011) including the update of reaction rate coefficients to the latest recommendations, Evaluation No. 18, of the Jet Propulsion Laboratory (Burkholder et al., 2015) and to values coming from other recent laboratory studies. A table of all affected reactions can be found in the Supplement (Tab. S1). Moreover, the yield of the photolysis of $CFCl_3$ (CFC-11) and $CF_2Cl_2$ (CFC-12) changed from three and two, respectively, to one chlorine (Cl) atom. The smaller Cl yield influences the $O_3$ mixing ratio in the stratosphere as Cl acts as a catalyst in the $O_3$ depleting cycles. The $O_3$ mixing ratio is higher everywhere in the stratosphere, except in the lowermost tropical stratosphere, in REF MLO compared to REF fSST (see Fig. S19). This results further in higher temperatures in the stratosphere in REF MLO (not shown). The contribution of the $ClO_x$ $O_3$ depleting cycle on total $O_3$ loss peaks at around 40 to 45 km altitude (see Fig. 5.28 in Seinfeld and Pandis, 2016). This corresponds approximately to the altitude of the maximum relative difference of $O_3$ mixing ratio between REF MLO and REF fSST (see Fig. S19).

## Appendix B

In the REF QFLX simulation the setting of the non-orographic gravity wave drag parameterization (GWAVE, Baumgaertner et al., 2013) was different than in the other simulations, in which breaking of gravity waves transfers only momentum, but no heat. In REF QFLX heat is also transferred leading to higher temperatures in the mesosphere. Since predominantly the mesosphere is affected, the different setting does not considerably influence the retrieved heat flux correction at the surface, the determination of which is the purpose of REF QFLX.

*Author contributions.* The simulations were set-up and carried out by PJ and FW with contributions of MK in applying the MLOCEAN submodel. MP and FW contrived and carried out the radiative impact and stratospheric adjusted temperature calculations and FW created the corresponding figures. LS analysed the data, created the remaining figures and prepared the manuscript with significant contributions regarding the interpretation and evaluation of the model results from all coauthors.

*Competing interests.* The authors declare that they have no conflict of interest.

*Acknowledgements.* We acknowledge the financial support by the DFG Project WI 5369/1-1 and the DLR internal projects KliSAW (Klimarelevanz von atmosphärischen Spurengasen, Aerosolen und Wolken) and MABAK (Innovative Methoden zur Analyse und Bewertung von Veränderungen der Atmosphäre und des Klimasystems). The model simulations have been performed at the German Climate Computing Centre (DKRZ) through support from the Bundesministerium für Bildung und Forschung (BMBF). We used the Climate Data Operators (CDO;





https://code.mpimet.mpg.de/projects/cdo) for data processing and the NCAR Command Language (NCL; https://doi.org/10.5065/D6WD3XH5)
for data analysis and to create the figures of this study. We furthermore thank all contributors of the project ESCiMo (Earth System Chem-
istry integrated Modelling), which provides the model configuration and initial conditions. We thank Roland Eichinger for his constructive
internal review of the manuscript and Hella Garny for her helpful comments on the interpretation of the dynamically induced temperature
response.





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
