# Peer review of "Slow Feedbacks Resulting from Strongly Enhanced Atmospheric Methane Concentrations in a Chemistry-Climate Model with Mixed Layer Ocean"

_Atmospheric Chemistry and Physics, 2020_

## Referee Comment (RC1) · Anonymous Referee #1 · 3 Aug 2020

The present manuscript by Stecher et al. investigates the climate impacts of 2x and 5x CH4 by simulations with the coupled chemistry-climate model EMAC. The study is an extension of the paper by Winterstein et al. (2019), who performed similar simulations, but with a focus on instantaneous impacts. To take into account slow climate feedbacks Stecher et al. ran the model with a mixed-layer ocean, while the previous simulations by Winterstein and colleagues used fixed sea surface temperatures and sea ice coverage as lower boundary condition, which limits the tropospheric response to increased methane concentrations. Overall, I think this is a solid model study which is of interest for a wider community. The paper is in principle well written. However, the paper refers very strongly to the previous study by Winterstein et al, and is thus extremely hard to

follow for readers, who are not familiar with the companion paper. Given the substantial length of the present manuscript not everybody is interested in reading a second paper in parallel. Therefore, I suggest major revisions and a substantial re-structuring of the paper before publication.

I have three major concerns:

1) As mentioned above the study is strongly linked to the work by Winterstein et al. Unfortunately, both studies have been conducted with different model versions. Moreover, the reference simulation for the MLO, REF QFLX, has been performed using a third model version / set-up. I have a hard time understanding why the authors did not simply apply the same model version as in Winterstein et al.? The authors want to make us believe that the model modifications do not have a significant impact on the outcome and try to circumvent this issue by showing differences of differences, which by the way does not increase readability, but how can you be sure that the climate background state has no impact on the modelled response to 2x (5x) CH4?

2) As already mentioned above the presentation of the results is strongly linked to the paper by Winterstein et al. Without knowing that paper, I find it often very difficult to follow the argumentation. For example, the paper discusses on increase in SWV due to enhanced atmospheric methane, but Fig. 6 displays negative changes in SWV as it present the difference SWV response in the MLO and fSST runs. This way of presenting the results is not very intuitive as the reader first has to look into the supplement to find the SWV response to enhanced CH4 in the MLO runs and then has to think about differences between the fSST and the MLO set-up. For the sake of readability and clarity I suggest to re-structure the paper as follows: First present the results of the MLO runs and move several of the figures provided in the supplement to the main paper, and then discuss the differences to Winterstein et al., maybe only for one case (2x or 5x), if the paper turns out to become too long.

3) The argumentation is often very qualitative, but not quantitative. A good example

is the discussion of SWV changes and their attribution to changes in CPT and CH4 oxidations. The CPT changes could be transferred into a change in H2O entry values, and from the model simulations it should be easy to calculate SWV production from CH4 oxidation. With that, the importance of both effects could be quantified. This is only one example, but there are several places where some more quantification would be desirable.

Specific comments:

- The title is very general, almost the same meaning as Winterstein et al.

- L6 and introduction: It would be nice to see a short definition/description of instantaneous and slow responses / feedbacks. Maybe it would be helpful to add a schematic to the paper, which shows the considered processes and clearly separates fast and slow effects.

- L90/91: I am bit confused by the description of the applied CH4 boundary condition. I thought that CH4 is relaxed towards to observational data set, and that this data set is simply multiplied by 2 (5) for the sensitivity runs. Why the "equilibrium CH4 fields of the respective fSST simulations"? What is the difference / advantage?

- L91 onwards: What is the advantage / difference between the relaxation approach and simply prescribing the CH4 concentration at the surface? What relaxation time scale is used? With the long lifetime of CH4 there should not be a large difference?

- REF QFLX: This simulation should be the same as REF fSST, shouldn't it? Does REF MLO also include the gravity wave set-up as described in Appendix B? If not, do you expect any impact?

- L127/128: What is the reason for the negative bias and observed total column CH4? This is a good example where an explanation seems to be given in Winterstein et al., but is unfortunately not summarized in the present study.

- L136/137, Fig. S1: I have also worked with the ECHAM5 MLO, and I am a bit concerned about the difference pattern shown in Fig. S1, namely the temperature difference around 60S, especially over the eastern hemisphere. In my simulation the MLO was in much better agreement with the reference SST climatology. Any thoughts about this?

- MLO: A more general question to the MLO: The MLO does not consider heat exchange with the deep ocean, but all forcing goes into the MLO. Up to which forcing strength is the usage of an MLO justified?

- L173 onwards: If Bony et al discussed a similar feature for CO2, then why not adding a short (speculative) discussion for CH4?

- L205-214: It would be nice to see some more quantification of the temperature effect on the CH4 lifetime!

- L232: Why is the tropospheric CH4 response marginally larger? Tropospheric is largely controlled by boundary condition? Remaining effect from CH4 oxidation?

- L246 onwards: Again the argumentation in this section stays mainly qualitative ("weaker increases of OH are presumably connected..."). Although the arguments sound reasonable, it should be possible to keep track of chemical production / loss budgets in a CCM.

- L293: Which one is the limiting OH precursor ? Water vapor or ozone? I would imagine that depends on the atmospheric region.

- L297: Please add a short summary of the explanation for the O3 response given in Winterstein et al.

- Fig. S9: Would be nice to see the difference in CPT for the reference simulations, fSST and MLO, as well

Technical comments:

- Eq (1): is there a bug in the listed units? E.g., units for reaction rate coefficient? [cm3

mol-1 s-1]? Otherwise the lifetime is not in [s].

- L190: [kg], to be consistent with the other units

- L324/325: It is not necessary to additionally mention numbers listed in a table here

---

## Referee Comment (RC2) · Peer Johannes Nowack (Referee) · 5 Aug 2020

Stecher et al. use a mixed-layer ocean chemistry-climate model to study the climate and atmospheric composition (ozone, water vapour) response to large idealised 2x and 5x methane forcings. The rapid adjustment response using fixed sea surface temperature (SST) simulations was reported elsewhere (Winterstein et al. 2019). The study here therefore focuses on the characteristics of the slow - surface warming mediated - climate feedback response. In particular, the authors discuss the factors driving changes in tropospheric methane lifetime, the non-linearity in the response with increasing methane forcing, the factors influencing stratospheric water vapour, as well

as the radiative impacts due to the associated changes in atmospheric composition. Finally, they also contrast radiatively and dynamically driven temperature changes.

This is a nice modelling study presenting several interesting and novel results. I agree with the second reviewer in the sense that sometimes references to the Winterstein paper could be reduced, or at least supported by additional figures and explanations. However, this should be possible to achieve with fairly straightforward adaptations. Below I list several suggestions for minor revisions subject to which I recommend rapid publication.

**Two thoughts on the wider context:**

- This work only considers the effects of increased methane in isolation, which is useful to separate its effect from those of other climate forcing agents. However, given the dependency of methane on, e.g., OH, I would expect that simultaneous CO2 forcing found in the real world could strongly interact with this picture, possibly even in a non-linear fashion. I assume that the reduction in OH driven by methane increases, for example, would be largely offset by increases in tropospheric OH under additional CO2 forcing? I am not asking that the study is revised in this sense, but the potential of such interactions should be mentioned somewhere, unless the authors can make strong arguments against this idea. A simple way to achieve this would be to add another clarifying sentence to the paragraph l. 204-214, where you discuss the importance of water vapour and ozone changes, which will also be driven by CO2 forcing and the associated tropospheric warming, thus impacting OH.

- Did the authors look at changes in the tropospheric circulation at all (cf. Chiodo Polvani 2016, Nowack et al. 2017)? I don't think any study has explored the specifics of the response to methane forcing, with its coupled effects on ozone and stratospheric water vapour before. I am NOT referring to the difference between the fixed SSTs and MLO runs here (Figure 2), as this might indeed be

beyond the scope of this work. If the model set-up allows (fairly short simulations and constrained ocean response), a short section on some central aspects of the tropospheric circulation response could further increase the impact of this paper. Otherwise, maybe suggest this point for future work with fully coupled ocean models. I could also imagine that the (lack of) tropospheric circulation changes might affect the stratospheric circulation response, e.g. through wave forcing and propagation, which might be worth commenting on.

- Chiodo & Polvani. Reduced Southern Hemispheric circulation response to quadrupled CO2, Geophysical Research Letters (2016).
- Nowack et al. On the role of ozone feedback in the ENSO amplitude response under global warming, Geophysical Research Letters (2017).

**Minor comments:**

- l. 6-8: it might be the passive use of verbs that makes this paragraph slightly hard to read, or also the reference to the Winterstein study. After all, all you seem to say is that: "Strong increases in CH4 reduce hydroxyl radical concentrations in the troposphere, thereby extending CH4 lifetime. We find that slow climate feedbacks counteract/dampen this effect (through increases in tropospheric water vapour and ozone(?); maybe mention the mechanism).

- l. 11-13: Maybe more explicitly say as well that the middle-upper stratospheric changes cannot be explained by changes in cold point temperature.

- l. 25: would rephrase "influenced". After all water vapour concentrations are also influenced anthropogenically, only is the effect indirect.

- l. 58-60: I am fairly sure that some of the NASA-GISS simulations by Drew Shindell might have had similar model set-ups but probably looked at other research questions?

- l. 85: Why not attempt a sensitivity analysis of the entire transient data following Gregory et al. GRL (2004) as well? Is the signal too small for the slope to be derived robustly? Gregory et al. A new method for diagnosing radiative forcing and climate sensitivity, Geophysical Research Letters (2004).

- l. 99: I suppose methane is not an emission flux then? Would be good to clarify to avoid misunderstandings.

- l. 139: One way of quantifying the importance of the climatological surface temperature differences would be to compare the global mean surface temperatures. I assume those differences should be smaller but possibly more relevant. Given that the MLO simulations are also free-running, could those effects also just represent some form of internal variability, which, if I understand correctly could still affect the sea ice distribution through atmospheric variability and its effect on SSTs? Higher latitudes can show similarly large variability for fully coupled ocean models. Similar arguments could apply to the NH (cf. l. 143). Looking at Fig. S1, I would think that the overall difference is positive, but the visual effect overemphasizes those changes in SH high latitudes which make up quite a small area. For climate sensitivity aspects, I would actually be more interested in the differences in tropical low-cloud regions which appear to stand out?

- l. 174/175: "would be beyond the scope"?

- l.180: It would indeed be useful to see the overall response, the rapid adjustment response and the difference due to slow feedbacks as subplots next to each other.

- l. 193: the tropopause is defined how?

- l. 228: revise sentence

- l. 258: you mean 'stratospheric abundance'

- Figure 6: another case where it would be useful to see the overall response as well instead of just the difference to the rapid adjustment response. Same for Figure 7. 2x2 panels?

- l. 334: the efficacy of ERF methane of close to 1 appears surprising to me – see e.g. the 145Hansen et al. Efficacy of climate forcings, Journal of Geophysical Research (2005).

- l. 368: how is this calculation of the effect on stratospheric temperatures done precisely? Could you provide more detail about the calculations? Are they expected to be robust in different regimes of the atmosphere, e.g. in the lowermost stratosphere vs the tropical upper stratosphere? What is "addst" in equation (2)?

---

## Author Comment (AC1) · 14 Oct 2020

**Reply to referee # 1**

October 14, 2020

Dear referee,

thank you very much for your comments and suggestions on our manuscript. In the following we reply to your comments point-by-point. The indicated pages of the answers relate to the discussion paper.

**1 Major concerns**

As mentioned above the study is strongly linked to the work by Winterstein et al. (2019). Unfortunately, both studies have been conducted with different model versions. Moreover, the reference simulation for the MLO, REF QFLX, has been performed using a third model version / set-up. I have a hard time understanding why the authors did not simply apply the same model version as in Winterstein et al. (2019)? The authors want to make us believe that the model modifications do not have a significant impact on the outcome and try to circumvent this issue by showing differences of differences, whichby the way does not increase readability, but how can you be sure that the climate background state has no impact on the modelled response to 2x (5x) methane ($CH_4$)?

We would like to clarify that the slow feedbacks can only be assessed as the shown difference of differences even if exactly the same model version was used. They are defined as the SST-driven contribution to the overall response and can therefore only be assessed as the difference of the overall response (as simulated in the MLO simulations) and the rapid adjustments (as simulated in the fSST simulations). We feel that we have not stated this clearly enough in the previous manuscript. We will state this in the introduction (see remark to line 6) and at the beginning of Sect. 3.3.1 (line 179 ff, see below).

The MLO simulations were performed at a later time than the fSST simulations. The submodel MLO-CEAN was not yet implemented in its full functionality in the model version used for the fSST simulations and backporting was not reasonable due to other changes in the model. Therefore, and also considering the computational cost of the simulations, we decided to run the MLO simulations with the most advanced cleanly defined model version available at that time.

Yes, we find differences in the simulated reference states of REF MLO and REF fSST (as shown for e.g. ozone ($O_3$) in Fig. S19), but these differences are small enough that they do not affect the conclusions about the differences between the full response and the rapid adjustments, i.e. the slow feedbacks.

**Old, l. 179 ff** Winterstein et al. (2019) analysed the quasi-instantaneous impact of doubled and fivefold $CH_4$ mixing ratios on the chemical composition of the atmosphere. In this section we investigate how tropospheric warming and associated climate feedbacks (see Sect. 3.2) modify these rapid adjustment patterns. For this purpose the difference patterns of the mixed layer ocean (MLO) sensitivity simulations are compared to those of the fSST simulations.

**New, l. 179 ff** Winterstein et al. (2019) analysed the quasi-instantaneous impact of doubled and fivefold $CH_4$ mixing ratios on the chemical composition of the atmosphere. In this section we investigate the respective slow feedbacks that are assessed as the difference between the full response (as simulated in the MLO simulations) and the rapid adjustments (as simulated in the fSST simulations) and therefore visualized as differences of the differences.

As already mentioned above the presentation of the results is strongly linked to the paper by Winterstein et al. (2019). Without knowing that paper, I find it often very difficult to follow the argumentation. For example, the paper discusses on increase in SWV due to enhanced atmospheric methane, but Fig. 6 displays negative changes in SWV as it present the difference SWV response in the MLO and fSST runs. This way of presenting the results is not very intuitive as the reader first has to look into the supplement to find the SWV response to enhanced $CH_4$ in the MLO runs and then has to think about differences between the fSST and the MLO set-up. For the sake of readability and clarity I suggest to re-structure the paper as follows: First present the results of the MLO runs and move several of the figures provided in the supplement to the main paper, and then discuss the differences to Winterstein et al. (2019), maybe only for one case (2x or 5x), if the paper turns out to become too long.

We understand that it is difficult to follow the presentation of the previous structure without knowing Winterstein et al. (2019). Therefore, we will make the following changes.

- We will add a short summary of the most important findings and conclusions of Winterstein et al. (2019) in the introduction (see remark to line 6). This implies that we do not need to refer to Winterstein et al. (2019) in the results section too often. We will review the results section with regard to this.

- We will show panel plots of the overall response (MLO) and slow feedbacks (difference between MLO and fSST) for temperature, hydroxyl radical (OH), water vapour ($H_2O$), and $O_3$. This should make it easier to interpret the slow response in comparison to the full response.

- We will also include a short description of the full response where we think it is necessary, e.g. for $O_3$. However, as the slow feedbacks represent only small modifications of the rapid adjustments, we think that it is not necessary to discuss the full response separated from the slow feedbacks and, that it would largely repeat the study of Winterstein et al. (2019).

The argumentation is often very qualitative, but not quantitative. A good example is the discussion of SWV changes and their attribution to changes in CPT and $CH_4$ oxidations. The CPT changes could be transferred into a change in H2O entry values, and from the model simulations it should be easy to calculate SWV production from $CH_4$ oxidation. With that, the importance of both effects could be quantified. This is only one example, but there are several places where some more quantification would be desirable.

We agree that more quantification would be desirable. Therefore, we went through the manuscript and have the following suggestions for improvement, first regarding the discussion about the $H_2O$ response:

- We had already included the relative change of $H_2O$ entry values in l. 273, but we will also include the absolute values in ppm as these might be more relevant. We have estimated the $H_2O$ entry value as the tropical (10°S–10°N) mean $H_2O$ mixing ratio at 70 hPa following Revell et al. (2016).

- The quantification of stratospheric water vapour (SWV) production from $CH_4$ oxidation is not so straightforward. It has often been assumed that two $H_2O$ molecules result from one oxidized $CH_4$ molecule, but Frank et al. (2018) showed that the yield deviates from two molecules and further varies with height. Tracing the chemical pathways to determine the actual yield of $H_2O$ is not so trivial and requires a comprehensive tagging mechanism (see also Frank et al., 2018). Another possibility is to estimate $H_2O$ from $CH_4$ oxidation as $H_2O_{CH4} = H_2O - H_2O_{entry}$. We have already done this qualitatively when we compared the change of $H_2O$ entry mixing ratio that is slightly higher in the MLO runs with the response of $H_2O$ in the middle and upper stratosphere that is lower in the MLO runs. We will calculate it explicitly with the formula above and include a Figure in the supplement.

**Old, l. 271 ff** The SWV mixing ratio at a given location and time can be approximated as the sum of these two terms (Austin et al., 2007; Revell et al., 2016). We calculate the amount of tropospheric $H_2O$ entering the stratosphere as the tropical (10°N–10°S) mean $H_2O$ mixing ratio at 70 hPa following Revell et al. (2016). The $H_2O$ entry mixing ratio increases by about 10 % (40 %) in the $CH_4$ doubling (fivefolding) experiments (both MLO and fSST). The relative increases are insignificantly higher in both MLO experiments compared to the respective fSST experiment. Furthermore, the zonal mean tropical cold point temperature (CPT) increases in all sensitivity simulations (see Fig. S9). The magnitude and the latitude dependence of the CPT changes are very similar for both doubling and both fivefolding experiments, although slightly larger for the MLO experiments in line with the changes of the $H_2O$ entry mixing ratio. Changes of the amount of tropospheric $H_2O$ entering the stratosphere can therefore not explain the differences in the SWV response between MLO and fSST in the middle and upper stratosphere. The increases of the $H_2O$ entry mixing ratio and the CPT are both slightly stronger in the MLO experiments and would therefore suggest a stronger increase of SWV in the MLO experiments. On the contrary, the increases of SWV are weaker in the middle and upper stratosphere in the MLO experiments compared to fSST. The contribution of the $CH_4$ oxidation on SWV can explain these weaker increases of SWV in the MLO experiments. The strengthening of the $CH_4$ oxidation in the stratosphere is weaker in the MLO experiments resulting likewise in a weaker increase of SWV produced by $CH_4$ oxidation.

**New, l. 271 ff** The SWV mixing ratio at a given location and time can be approximated as the sum of these two terms following Austin et al. (2007); Revell et al. (2016) as

$$H_2O = H_2O_{\text{entry}} + H_2O_{\text{CH4}}.$$

We calculate the amount of tropospheric $H_2O$ entering the stratosphere as the tropical (10°S–10°N) mean $H_2O$ mixing ratio at 70 hPa following Revell et al. (2016). The $H_2O$ entry mixing ratio increases by 9.08 % (0.14 ppm) in S2 fSST, 9.77 % (0.17 ppm) in S2 MLO, 38.53 % (0.57 ppm) in S5 MLO, and 38.86 % (0.68 ppm) in S5 MLO. Furthermore, the zonal mean tropical CPT increases in all sensitivity simulations (see Fig. S9). Though differences exist between the reference CPT in MLO und fSST, the magnitude and latitudinal structure of the CPT changes are very similar for both doubling and both fivefolding experiments. They are also a bit larger for the MLO experiments (again consistent for the S2 and S5 case), in line with the response of the $H_2O$ entry mixing ratios. Changes of the amount of tropospheric $H_2O$ entering the stratosphere can therefore not explain the weaker increase of SWV in the MLO experiments compared to fSST in the middle and upper stratosphere.

To illustrate the effect of $CH_4$ oxidation on the SWV response, Fig. S8 shows the response of $H_2O$ from $CH_4$ oxidation estimated using Eq. 2. As discussed in the previous paragraph, the strengthening of the $CH_4$ oxidation in the stratosphere is weaker in the MLO experiments. This results in a weaker increase of SWV produced by $CH_4$ oxidation in the middle and upper stratosphere (see Fig. S8 c) d)) and can explain the difference of SWV response between MLO and fSST as shown in Fig. 6.

> In addition, we will also include the following points:
>
> - We will add the tropospheric $CH_4$ lifetime when only the temperature dependent reaction rate coefficient responds to the forcing (see remark to line 205–214).
>
> - To quantify the composition changes in the tropical lower stratosphere we will give average values of $CH_4$ and $O_3$ changes in boxes in this region.

For $CH_4$:

**Old, l. 238 ff** Another aspect to note in Fig. 5 is the more than $5\times CH_4$ increase in the lowermost tropical stratosphere for S5 MLO. This feature indicates enhanced tropical upwelling, which leads to larger $CH_4$ mixing ratios in the tropical lower stratosphere. This feature is more pronounced in S5 MLO than in S5 fSST, in line with the more pronounced changes of tropical upwelling in the MLO set-up as discussed in Sect. 3.2.

**New, l. 238 ff** Another aspect to note in Fig. 5 is the more than $2\times$ or $5\times CH_4$ increase in the lowermost tropical stratosphere. This feature indicates enhanced tropical upwelling, which leads to larger $CH_4$ mixing

ratios in the tropical lower stratosphere. It is more pronounced in the MLO than in the fSST experiments, in line with the more pronounced changes of tropical upwelling in the MLO set-up as discussed in Sect. 3.2. The average deviation from $2\times$ or $5\times CH_4$ for a region in the tropical lower stratosphere (30°S–30°N, 70–20 hPa) is 0.16 % for S2 fSST, 0.37 % for S2 MLO, 0.23 % for S5 fSST, and 1.31 % for S5 MLO.

For $O_3$:

**Old, l. 298** A dominant feature is the stronger decrease of $O_3$ in the lowermost tropical stratosphere in S5 MLO compared to S5 fSST of up to 18 percentage points (p.p.). This difference also exists between the S2 simulations, albeit weaker (4 p.p.).

**New, l. 298** A dominant feature is the stronger decrease of $O_3$ in the lowermost tropical stratosphere in S5 MLO compared to S5 fSST of up to 18.39 p.p.. The average difference between S5 MLO and S5 fSST for a region in the tropical lower stratosphere (30°S–30°N, 100–20 hPa) is 6.33 p.p.. This difference also exists between the S2 simulations, albeit weaker (with a maximum difference of 4.68 p.p. and an average difference of 1.67 p.p.).

**2 Specific comments**

The title is very general, almost the same meaning as Winterstein et al. (2019).

We will change the title to *Slow Feedbacks Resulting from Strongly Enhanced Atmospheric Methane Concentrations in a Chemistry-Climate Model with Mixed Layer Ocean* to emphasize that this study focuses on the slow SST-driven feedbacks.

L6 and introduction: It would be nice to see a short definition/description of instantaneous and slow responses / feedbacks. Maybe it would be helpful to add a schematic to the paper, which shows the considered processes and clearly separates fast and slow effects.

While we think that the key parameters of the conceptual radiative forcing, radiative feedback, and climate sensitivity framework adopted here, have all been mentioned and defined in the original manuscript, we admit that the referee's proposal of a compact presentation in the introduction is certainly worthwhile. To account for the referee's request, we have reorganized and somewhat extended the introduction, starting at l.42. However, since our manuscript already contains a lot of Figures, we tend to not include an additional schematic.

**Old, l. 42** However, these studies did not focus on the climate impact of $CH_4$. Other recent studies assessing climate feedbacks and climate sensitivity of $CH_4$ did not include radiative contributions from chemical feedbacks in their analysis (Modak et al., 2018; Smith et al., 2018; Richardson et al., 2019).

Winterstein et al. (2019) assessed chemical feedback processes and their radiative impact (RI) in sensitivity simulations forced by 2-fold ($2\times$) and 5-fold ($5\times$) present-day (year 2010) $CH_4$ mixing ratios. As their simulation set-up prescribed sea surface temperatures (SSTs) and sea ice concentrations (SICs) and thus suppressed surface temperature changes, the parameter changes in their simulations have the character of rapid adjustments (e.g., Forster et al., 2016; Smith et al., 2018). In the effective radiative forcing (ERF) framework, rapid adjustments of radiatively active species are counted as part of the forcing and are to be distinguished from slow climate feedbacks that are coupled to surface temperature changes (Sherwood et al., 2015). Climate sensitivity parameters, reflecting the degree of surface temperature change per unit forcing, have been found to be less dependent on the forcing agent with this definition compared to previous definitions of radiative forcing (RF) (e.g., Shine et al., 2003; Hansen et al., 2005; Richardson et al., 2019).

As a follow-up on Winterstein et al. (2019), we assess the respective SST-driven climate feedbacks, their effect on the quasi-instantaneous response of the chemical composition, and consequently resulting radiative feedbacks. Consistent with Winterstein et al. (2019), we perform sensitivity simulations with $2\times$ and $5\times$ present-day $CH_4$ mixing ratios with the ECHAM/MESSy Atmospheric Chemistry (EMAC) chemistry-climate model (CCM) (Jöckel et al., 2016), but this time coupled to a MLO model instead of prescribing SSTs and SICs. To our knowledge, this is the first study assessing the response to strong increases of $CH_4$ mixing ratios in a fully coupled CCM, meaning that the interactive model system includes atmospheric dynamics, atmospheric chemistry, and ocean thermodynamics.

**New, l. 42** However, these studies did not focus on the climate impact of $CH_4$. In climate feedback and sensitivity studies it has become standard to distinguish between rapid adjustments of the system (that develop in direct reaction to the forcing, independently from sea surface temperature changes) and feedbacks driven by slowly evolving temperature changes at the Earth's surface (e.g., Colman and McAvaney, 2011; Geoffroy et al., 2014; Smith et al., 2020). Under this concept, the rapid radiative adjustments are counted as an integral part of the radiative forcing, yielding the so-called effective radiative forcing (Shine et al., 2003; Hansen et al., 2005). The concept has been found to be physically more meaningful than other radiative forcing frameworks, as the climate sensitivity parameter, i.e., the global mean surface temperature change per unit radiative forcing, is becoming less dependent on the forcing agent (Hansen et al., 2005; Sherwood et al., 2015; Richardson et al., 2019). However, recent studies of climate feedbacks and sensitivity to a $CH_4$ forcing adopting the effective radiative forcing concept did not account for the radiative contribution from chemical feedbacks in their analysis (Modak et al., 2018; Smith et al., 2018; Richardson et al., 2019).

Winterstein et al. (2019) assessed chemical feedback processes and their RI in simulations forced by 2-fold ($2\times$) and 5-fold ($5\times$) present-day (year 2010) $CH_4$ mixing ratios. As their simulation set-up used prescribed sea surface temperatures (SSTs) and sea ice concentrations (SICs) and thus suppressed surface temperature changes, the parameter changes in their simulations match the rapid adjustment and effective radiative forcing concept (e.g., Forster et al., 2016; Smith et al., 2018). Rapid radiative adjustments to stratospheric ozone and water vapor changes were found to make a considerable contribution to the $CH_4$ effective radiative forcing, in line with previous respective findings (e.g., Shindell et al., 2005, 2009; Stevenson et al., 2013). SWV mixing ratios were found to increase steadily with height under increased $CH_4$ in the quasi-instantaneous response as analysed by Winterstein et al. (2019). Rapid adjustments of the chemical composition of the stratosphere lead to increases of OH favoring the depletion of $CH_4$, which is an important in situ source of SWV. The increased SWV mixing ratios cool the stratosphere, thereby affecting $O_3$. In the troposphere, the enhanced $CH_4$ burden leads to a strong reduction of its most important sink partner, OH, thereby affecting the $CH_4$ lifetime. Winterstein et al. (2019) found a near-linear prolongation of the tropospheric $CH_4$ lifetime with increasing scaling factor of $CH_4$ for the two conducted experiments ($2\times$ and $5\times CH_4$).

As a follow-up on Winterstein et al. (2019), we assess the respective slow SST-driven response of the chemical composition and resulting radiative feedbacks. Consistent with Winterstein et al. (2019), we perform sensitivity simulations with $2\times$ and $5\times$ present-day $CH_4$ mixing ratios with the EMAC CCM (Jöckel et al., 2016), but this time coupled to a MLO model instead of prescribing SSTs and SICs. For radiative forcing strengths as discussed here, equilibrium climate sensitivity simulations using a thermodynamic mixed layer ocean as lower boundary condition have been shown to represent the surface temperature response yielded in (much more resource demanding) model setups involving a dynamic deep ocean sufficiently well (e.g., Danabasoglu and Gent, 2009; Dunne et al., 2020; Li et al., 2013). The slow feedbacks are assessed as the difference between the full response (as simulated in the MLO simulations) and the rapid adjustments (as simulated in the simulations with prescribed SSTs and SICs). To our knowledge, this is the first study assessing the response to strong increases of $CH_4$ mixing ratios in a fully coupled CCM, meaning that the interactive model system includes atmospheric dynamics, atmospheric chemistry, and ocean thermodynamics.

L90/91: I am bit confused by the description of the applied $CH_4$ boundary condition. I thought that $CH_4$ is relaxed towards to observational data set, and that this data set is simply multiplied by 2 (5) for the sensitivity runs. Why the "equilibrium $CH_4$ fields of the respective fSST simulations"? What is the

difference / advantage?

We apply the nudging of the $CH_4$ mixing ratio to the observational data set only at the lower boundary. The atmospheric $CH_4$ mixing ratios are free to adjust to this forcing. In the stratosphere, for example, the increase of $CH_4$ mixing ratio deviates from the increase factors of 2 and 5, respectively.

As the equilibrium fields of $CH_4$ mixing ratio from the fSST experiments are already close to the respective equilibrium of the MLO simulations, the initialization with these fields shortens the spin-up. We will reformulate the sentence to state this point more clearly.

**Old** The MLO simulations have been initialized with the equilibrium $CH_4$ fields of the respective fSST simulations, thus the initial $CH_4$ fields of S2 MLO and S5 MLO were implicitly scaled by two and five, respectively.

**New** The MLO simulations have been initialized with the equilibrium $CH_4$ fields of the respective fSST simulations. As the latter are already close to the respective equilibrium $CH_4$ fields of the MLO simulations, the initialization with these fields shortens the spin-up.

L91 onwards: What is the advantage / difference between the relaxation approach and simply prescribing the $CH_4$ concentration at the surface? What relaxation timescale is used? With the long lifetime of $CH_4$ there should not be a large difference?

Indeed, it is in principle the same as the relaxation time (10800 s) is short in comparison with the $CH_4$ lifetime and transport times. We will add the nudging coefficient to the manuscript.

**Old** Alike the fSST simulations, the $CH_4$ lower boundary mixing ratios of the MLO simulations are prescribed by Newtonian relaxation (i.e. nudging).

**New** Alike the fSST simulations, the $CH_4$ lower boundary mixing ratios of the MLO simulations are prescribed by Newtonian relaxation (i.e. nudging) with a nudging coefficient of 10800 s.

REF QFLX: This simulation should be the same as REF fSST, shouldn't it? Does REF MLO also include the gravity wave set-up as described in Appendix B? If not, do you expect any impact?

In principle, REF QFLX should be the same as REF fSST, but the simulations were performed with different model versions. ALL MLO simulations use the same gravity wave set-up as the fSST simulations for consistency. The different gravity wave set-up does mainly influence the middle atmosphere. We therefore presume that the influence on the ground is so small that the heat flux correction is not affected.

**Old** In the REF QFLX simulation the setting of the non-orographic gravity wave drag parameterization (GWAVE, Baumgaertner et al., 2013) was different than in the other simulations, ...

**New** In the REF QFLX simulation the setting of the non-orographic gravity wave drag parameterization (GWAVE, Baumgaertner et al., 2013) was different than in all the other simulations (fSST and MLO), ...

L127/128: What is the reason for the negative bias and observed total column $CH_4$? This is a good example where an explanation seems to be given in Winterstein et al. (2019), but is unfortunately not

summarized in the present study.

Thank you for this note. We will add a short explanation to the text.

**Old, l. 127 ff** Consistent with REF fSST (see Winterstein et al., 2019), there is a negative bias between the REF MLO and the observed total $CH_4$ columns of less than 4 % (not shown). Given that relative comparisons between sensitivity simulations and the reference are the main target of our analysis, REF MLO represents $CH_4$ conditions of the year 2010 sufficiently realistic for our purpose.

**New, l. 127 ff** Consistent with REF fSST (see Winterstein et al., 2019), there is a negative bias between the REF MLO and the observed total $CH_4$ columns of less than 4 % (not shown). Note that not all the observations originate precisely from the year 2010. The global annual mean $CH_4$ surface mixing ratios have, for example, risen by about 0.024 ppm from 2010 to 2014 (www.esrl.noaa.gov/gmd/ccgg/trends_ch4/), the year of the study by Klappenbach et al. (2015). In addition, the $CH_4$ lifetime could be slightly underestimated. The $CH_4$ lifetime in EMAC lies in the middle to lower range in comparisons with other CCMs (Jöckel et al., 2006; Voulgarakis et al., 2013). However, given that relative comparisons between sensitivity simulations and the reference are the main target of our analysis, REF MLO represents $CH_4$ conditions of the year 2010 sufficiently realistic.

L136/137, Fig. S1: I have also worked with the ECHAM5 MLO, and I am a bit concerned about the difference pattern shown in Fig. S1, namely the temperature difference around 60S, especially over the eastern hemisphere. In my simulation the MLO was in much better agreement with the reference SST climatology. Any thoughts about this?

We have derived the flux correction at the surface that stabilizes the MLO reference run from the surface fluxes of the fixed SST reference run. If you did likewise in your coupling exercise, one possibility could be that your basic model (with fixed SSTs) has had an ideally balanced top of the atmosphere radiation balance, with optimally low correction fluxes. (In our case the original global radiation balance was -1.14 Watt per square meter ($W\,m^{-2}$) for REF fSST.) As the largest temperature deviations occur near the ice edge, another possibility could be that you provided a multiple iteration of the correction fluxes in these regions to ensure optimal reproduction of the ice edge location in the reference run with MLO. Did you?

MLO: A more general question to the MLO: The MLO does not consider heat exchange with the deep ocean, but all forcing goes into the MLO. Up to which forcing strength is the usage of an MLO justified?

We feel that there is robust, long standing evidence for a sufficient reproduction of climate sensitivity parameters simulated by deep ocean coupled AOGCMs by MLO coupled AOGCMs in case of forcing strengths at least up to carbon dioxide ($CO_2$) doubling (Danabasoglu and Gent, 2009; Dunne et al., 2020). This evidence has been explicitly confirmed for the ECHAM5 climate model (Li et al., 2013), which is the atmospheric model basic to the chemistry-climate model setup used in our paper. Problems may arise for larger forcings ($4xCO_2$ and higher) with strong ocean mixed layer warming, which is not transferred to the deep layers, but as our forcings are much smaller than for $CO_2$ doubling, that should not be an issue here.
We will add a clarifying sentence to the paragraph introducing the MLO setup (l. 58, see also remark to line 6):

**Old, l. 58** ... of prescribing SSTs and SICs.

**New, l. 58** ... of prescribing SSTs and SICs. For radiative forcing strengths as discussed here, equilibrium climate sensitivity simulations using a thermodynamic mixed layer ocean as lower boundary condition have been shown to represent the surface temperature response yielded in (much more resource demanding) model setups involving a dynamic deep ocean sufficiently well (e.g., Danabasoglu and Gent, 2009; Dunne et al., 2020; Li et al., 2013).

> L173 onwards: If Bony et al discussed a similar feature for $CO_2$, then why not adding a short (speculative) discussion for $CH_4$?

> Our previous formulation was a bit vague. What we wanted to indicate is the following: Bony et al. (2013) found differences between the fast and the slow (temperature driven) response of the tropospheric tropical circulation in $CO_2$ increase experiments. We will state that more clearly.
> However, we still think that a detailed discussion of the processes leading to these differences is beyond the scope of this paper. As proposed by the second referee, Peer Nowack, we will add an outlook on tropospheric circulation changes in $CH_4$ increase simulations as this is surely an interesting research question by itself.

**Old, l. 173 ff** A similar feature has been noticed and discussed in $CO_2$ increase simulations, too (e.g. Bony et al., 2013). However, ...

**New, l. 173 ff** Differences between the fast and the slow response of the tropospheric tropical circulation have been noticed and discussed in $CO_2$ increase simulations, too (e.g. Bony et al., 2013). However, ...

> L205-214: It would be nice to see some more quantification of the temperature effect on the $CH_4$ lifetime!

> Fig. 3 shows the total effect on the $CH_4$ lifetime that results from changes of $CH_4$, OH and the temperature dependent reaction rate coefficient. A possible quantification of the temperature effect on $CH_4$ lifetime would be the comparison with the $CH_4$ lifetime calculated using only a changed reaction rate coefficient corresponding to temperatures of $2\times$ and $5\times$ $CH_4$. However, also the abundance of OH is influenced by temperature changes as we show in this study. Therefore, changing only the reaction rate coefficient would not represent the whole temperature/climate effect on the $CH_4$ lifetime.
> Nevertheless, we will include the isolated effect of the temperature dependent reaction rate on the $CH_4$ lifetime in Fig. 3.

**Old, l. 205 ff** Additionally, the tropospheric warming in the MLO sensitivity simulations results in a faster $CH_4$ oxidation as its reaction rate increases with temperature.

**New, l. 205 ff** Additionally, the tropospheric warming in the MLO sensitivity simulations results in a faster $CH_4$ oxidation as its reaction rate increases with temperature. The isolated effect of the temperature dependent reaction rate is indicated by the blue squares in Fig. 3. They show the $CH_4$ lifetime corresponding to REF MLO, except for the reaction rate coefficient that was calculated with temperatures corresponding to $2\times$ and $5\times$ $CH_4$.

> L232: Why is the tropospheric $CH_4$ response marginally larger? Tropospheric is largely controlled by boundary condition? Remaining effect from $CH_4$ oxidation?

> Yes, we think you are absolutely right. We will reorganize the paragraph to state this more clearly.

**Old, l. 232 ff** Winterstein et al. (2019) investigated whether the increase of atmospheric $CH_4$ follows the doubling or fivefolding for fSST conditions linearly. Tropospheric $CH_4$ is largely controlled by the nudging at the lower boundary through mixing and responds linearly to the increase. However, the $CH_4$ increase between 50 and 1 hPa has found to be smaller than a strictly linear relation would predict. This indicates enhanced chemical $CH_4$ depletion in the stratosphere due to changes in the chemical composition. Fig. **??** shows the relative difference between the annual zonal mean $CH_4$ of S2 MLO (S5 MLO) and $2\times$ ($5\times$) the zonal mean $CH_4$ of REF MLO. The doubling or fivefolding of the reference $CH_4$ serves to emphasize regions where the increase factor of the $CH_4$ mixing ratio deviates from 2 or 5, respectively. The response of tropospheric $CH_4$ is marginally larger than a linear increase in both MLO experiments. This is in line with the response of tropospheric $CH_4$ in the fSST simulations. As for the fSST simulations, the $CH_4$ increase in the extratropical stratosphere is weaker than a linear increase in both MLO sensitivity simulations. The non-linearity is less pronounced in the two MLO sensitivity experiments compared to the respective fSST experiments (compare with Fig. 3 in Winterstein et al., 2019) suggesting that the chemical depletion of $CH_4$ is enhanced in the MLO experiments as well, however, less strongly than in the fSST experiments.

**New, l. 232 ff** Fig. 5 shows the relative differences between the annual zonal mean $CH_4$ of S2 MLO (S5 MLO) and $2\times$ ($5\times$) the zonal mean $CH_4$ of REF MLO. The doubling or fivefolding of the reference $CH_4$ serves to emphasize regions where the increase factor of the $CH_4$ mixing ratio deviates from 2 or 5, respectively. The response of tropospheric $CH_4$ is marginally larger than a linear increase in both MLO experiments. This is in line with the response of tropospheric $CH_4$ in the fSST simulations. Tropospheric $CH_4$ is largely controlled by the nudging at the lower boundary through mixing and is, therefore, prevented to adjust to the lifetime increase as discussed above. The slightly positive values in Fig. 5 indicate a small residual of this effect. As for the fSST simulations, the $CH_4$ increase between 50 and 1 hPa is smaller than the factors of 2 or 5, respectively. This effect is less pronounced in the two MLO sensitivity experiments compared to the respective fSST experiments (compare with Fig. 3 in Winterstein et al., 2019) suggesting that the chemical depletion of $CH_4$ is enhanced in the MLO experiments as well, however, less strongly than in the fSST experiments.

> L246 onwards: Again the argumentation in this section stays mainly qualitative ("weaker increases of OH are presumably connected..."). Although the arguments sound reasonable, it should be possible to keep track of chemical production / loss budgets in a CCM

> Unfortunately, it is not trivial to keep track of the chemical production and loss budgets of OH in a comprehensive chemical mechanism such as MECCA. It is theoretically possible, but would require a complex tagging mechanism as presented by, e.g., Gromov et al. (2010). In the present simulations we did not use this mechanism as it is computationally expensive and can, therefore, not be applied to global simulations that cover multiple decades. For simple mechanisms, as for example the $CH_4$ sink reactions, keeping track of the budget is straightforward.

> L293: Which one is the limiting OH precursor? Water vapor or ozone? I would imagine that depends on the atmospheric region?

> As already replied to the previous remark, it is not easy to determine the production and loss budgets of OH from our simulation results. Determining the more important OH precursor is also not straightforward and would require additional calculations. Nicely et al. (2020), for example, assessed the contribution of various drivers to the $CH_4$ lifetime long-term trend (as proxy for OH) with a machine learning algorithm.
>
> Here, we can only speculate if $H_2O$ or $O_3$ is the limiting precursor for stratospheric OH. Our reasoning here is that, as the increase in OH is smaller in the MLO runs, while the entry of tropospheric $H_2O$ is stronger, the limiting precursor is presumably $O_3$.

L297: Please add a short summary of the explanation for the $O_3$ response given in Winterstein et al. (2019).

We will add a short summary of the explanation for the $O_3$ response.

**Old, l. 296 ff** Winterstein et al. (2019) gave a detailed explanation of the processes leading to the resulting $O_3$ pattern that is also valid for the MLO simulations.

**New, l. 296 ff** Winterstein et al. (2019) gave a detailed explanation of the processes leading to the resulting $O_3$ pattern that is also valid for the MLO simulations. As the $O_3$ catalytic depletion cycles are less efficient at lower temperatures radiative cooling in the stratosphere results in increased $O_3$ mixing ratios in the middle stratosphere (between 50 and 5 hPa). Additionally, increased abundances of $H_2O$ favor the depletion of excited oxygen ($O(^1D)$), likewise reducing the sink of $O_3$ and favoring increases of the $O_3$ abundance. Reduced $O_3$ mixing ratios in the lowermost tropical stratosphere indicate enhanced tropical upwelling of $O_3$ poor air from the troposphere into the stratosphere. Above 2 hPa, increases of OH lead to enhanced depletion of $O_3$ resulting in reduced $O_3$ mixing ratios.

Fig. S9: Would be nice to see the difference in CPT for the reference simulations, fSST and MLO, as well.

We will include the difference of cold point temperature of REF MLO and REF fSST in Fig. S9.
In addition, we will make the following change to the manuscript (see also answer to major concern 3)).

**Old, l. 276 ff** The magnitude and the latitude dependence of the CPT changes are very similar for both doubling and both fivefolding experiments, although slightly larger for the MLO experiments in line with the changes of the $H_2O$ entry mixing ratio.

**New, l. 276 ff** Though differences exist between the reference CPT in MLO und fSST, the magnitude and latitudinal structure of the CPT changes are very similar for both doubling and both fivefolding experiments. They are also a bit larger for the MLO experiments (again consistent for the S2 and S5 case), in line with the response of the $H_2O$ entry mixing ratios.

**3 Technical corrections**

**Page 7, line 189, Equation (1):** is there a bug in the listed units? E.g., units for reaction rate coefficient? [cm3 mol-1 s-1]? Otherwise the lifetime is not in [s].

**Page 7, line 190:** [kg], to be consistent with the other units.

**Pages 14-15, lines 324-325:** It is not necessary to additionally mention numbers listed in a table here.

Thank you for these suggestions and corrections. We fully agree and changed the manuscript accordingly. The unit of the concentration in Eq. (1) is [cm-3].

---

## Author Comment (AC2) · 14 Oct 2020

**Reply to referee # 2**

October 14, 2020

Dear Peer Johannes Nowack,

thank you very much for the positive comments on our manuscript. In the following we reply to your comments point-by-point. The indicated pages of the answers relate to the discussion paper.

**1   Thoughts on the wider context**

> This work only considers the effects of increased methane in isolation, which is useful to separate its effect from those of other climate forcing agents. However, given the dependency of methane on, e.g., OH, I would expect that simultaneous CO2 forcing found in the real world could strongly interact with this picture, possibly even in a non-linear fashion. I assume that the reduction in OH driven by methane increases, for example, would be largely offset by increases in tropospheric OH under additional CO2 forcing? I am not asking that the study is revised in this sense, but the potential of such interactions should be mentioned somewhere, unless the authors can make strong arguments against this idea. A simple way to achieve this would be to add another clarifying sentence to the paragraph l. 204-214, where you discuss the importance of water vapour and ozone changes, which will also be driven by CO2 forcing and the associated tropospheric warming, thus impacting OH.

> Thank you for making this point. We fully agree and will rephrase the paragraph as follows.

**Old, l. 212** ... century. However, the tropospheric warming in the RCP8.5 scenario is stronger because it includes the effects of all greenhouse gass (GHGs) and not only the effect of methane ($CH_4$). This can explain the larger offset of the $CH_4$ lifetime response reported by Voulgarakis et al. (2013).

**New, l. 212** ... century. However, the tropospheric warming in the RCP8.5 scenario is stronger because it includes the effects of all GHGs, as opposed to the isolated effect of $CH_4$ in our experiments. Additional warming induced by other GHGs, in particular carbon dioxide ($CO_2$), would drive water vapour ($H_2O$) and ozone ($O_3$) increases as well. Therefore, the reduction in hydroxyl radical (OH) driven by $CH_4$ increases in our experiments is expected to be more strongly offset under a simultaneously active $CO_2$ forcing.

> Did the authors look at changes in the tropospheric circulation at all (cf. Chiodo Polvani 2016, Nowack et al. 2017)? I don't think any study has explored the specifics of the response to methane forcing, with its coupled effects on ozone and stratospheric water vapour before. I am NOT referring to the difference between the fixed SSTs and MLO runs here (Figure 2), as this might indeed beyond the scope of this work. If the model set-up allows (fairly short simulations and constrained ocean response), a short section on some central aspects of the tropospheric circulation response could further increase the impact of this paper. Otherwise, maybe suggest this point for future work with fully coupled ocean models. I could also imagine that the (lack of) tropospheric circulation changes might affect the stratospheric circulation response, e.g. through wave forcing and propagation, which might be worth commenting on

The tropospheric circulation in response to $CH_4$ forcing with and without interactive chemistry would be a very interesting research question, indeed. However, we think that it would open up a new subject area. Considering that this paper is already quite long, we think that a discussion about tropospheric circulation changes is beyond the scope of the present paper and we prefer to leave this point for future work. Moreover, in a future study we plan to use a $CH_4$ emission flux boundary condition, as opposed to the prescribed $CH_4$ surface mixing ratios here, so that tropospheric $CH_4$ can adjust to changes in its sinks. We will include a suggestion of the topic for this study in the conclusions section.

**Old, l. 460** The contribution of sea surface temperature (SST)-driven climate feedbacks to the total $CH_4$ induced $O_3$ response shows remarkable similarities to the $O_3$ response to climate feedbacks in $CO_2$-forced climate change simulations (Dietmüller et al., 2014; Nowack et al., 2018; Chiodo and Polvani, 2019). The consistency between the $O_3$ feedbacks resulting from these different forcing agents encourages the separation of the $O_3$ response patterns into rapid adjustments and climate feedbacks in future studies. Rapid adjustments are specific to the forcing, whereas climate feedbacks are driven by surface temperature changes and are therefore expected to be less dependent on the forcing agent (Sherwood et al., 2015).

**New, l. 460** The contribution of SST-driven climate feedbacks to the total $CH_4$ induced $O_3$ response shows remarkable similarities to the $O_3$ response to climate feedbacks in $CO_2$-forced climate change simulations (Dietmüller et al., 2014; Nowack et al., 2018; Chiodo and Polvani, 2019). The consistency between the $O_3$ feedbacks resulting from these different forcing agents encourages the separation of the $O_3$ response patterns into rapid adjustments and climate feedbacks in future studies. Rapid adjustments are specific to the forcing, whereas climate feedbacks are driven by surface temperature changes and are therefore expected to be less dependent on the forcing agent (Sherwood et al., 2015). However, the overall response of $O_3$ (rapid adjustments and slow feedbacks) is quite different under $CH_4$ forcing compared to $CO_2$ forcing owing to chemically induced feedbacks under $CH_4$ forcing. Chiodo and Polvani (2017); Nowack et al. (2017) suggested that feedbacks from interactive $O_3$ under $CO_2$ forcing have the potential to significantly alter the tropospheric circulation. As the overall $O_3$ response is different under $CH_4$ forcing, also modified feedbacks on the tropospheric circulation are expected. Those are planned to be assessed using a simulation set-up with a $CH_4$ emission flux boundary condition to simulate feedbacks of tropospheric $CH_4$ to changes in its chemical sinks.

**2 Minor comments**

l. 6-8: it might be the passive use of verbs that makes this paragraph slightly hard to read, or also the reference to the Winterstein et al. (2019) study. After all, all you seem to say is that: "Strong increases in CH4 reduce hydroxyl radical concentrations in the troposphere, thereby extending CH4 lifetime. We find that slow climate feedbacks counteract/dampen this effect (through increases in tropospheric water vapour and ozone(?); maybe mention the mechanism).

Thank you for this suggestion. We will modify the text as follows.

**Old, l. 6** We find that the slow climate feedbacks counteract the reduction of the hydroxyl radical in the troposphere, which is caused by the strongly enhanced $CH_4$ mixing ratios. Thereby also the resulting prolongation of the tropospheric $CH_4$ lifetime is weakened compared to the quasi-instantaneous response considered previously.

**New, l. 6** Strong increases of $CH_4$ lead to a reduction of the hydroxyl radical in the troposphere, thereby extending the $CH_4$ lifetime. Slow climate feedbacks counteract this reduction of OH through increases in tropospheric $H_2O$ and $O_3$, thereby dampening the extension of $CH_4$ lifetime in comparison with the quasi-instantaneous response.

l. 11-13: Maybe more explicitly say as well that the middle-upper stratospheric changes cannot be explained by changes in cold point temperature

Thank you for this hint. We will change the text as follows.

**Old, l. 11** In the middle and upper stratosphere, the increase of stratospheric water vapour is reduced with respect to the quasi-instantaneous response. Weaker increases of the hydroxyl radical cause the chemical depletion of $CH_4$ to be less strongly enhanced and thus the in situ source of stratospheric water vapour as well.

**New, l. 11** In the middle and upper stratosphere, the increase of stratospheric water vapour is reduced with respect to the quasi-instantaneous response. We find that this difference cannot be explained by the response of the cold point and the associated $H_2O$ entry values, but by a weaker strengthening of the in situ source of $H_2O$ through $CH_4$ oxidation.

l. 25: would rephrase "influenced". After all water vapour concentrations are also influenced anthropogenically, only is the effect indirect.

Yes, this is indeed not correct. We will replace it by "directly emitted by human activity".

l. 58-60: I am fairly sure that some of the NASA-GISS simulations by Drew Shindell might have had similar model set-ups but probably looked at other research questions?

Thank you for this note. You are right, the work of Shindell et al. (2005, 2009) and Stevenson et al. (2013) should be mentioned here. We generally extended the introduction and also included these citations (see also reply to referee 1).
In addition, we will include the citation of Shindell et al. (2009) and Stevenson et al. (2013) when referring to Fig. 8.17 of the IPCC report: e.g., Fig. 8.17 in IPCC, 2013 derived from Shindell et al., 2009; Stevenson et al., 2013.

l. 85: Why not attempt a sensitivity analysis of the entire transient data following Gregory et al. GRL (2004) as well? Is the signal too small for the slope to be derived robustly? Gregory et al. A new method for diagnosing radiative forcing and climate sensitivity, Geophysical Research Letters (2004)

The signal is indeed too small for the slope to be derived robustly. We have actually tried this method and included Fig. 1 exemplary for the 5xCH4 case in this reply. For the 2xCH4 case, the signal to noise ratio is even worse.
One solution to reduce the uncertainty would be to calculate an ensemble of spin-up phases as proposed by, e.g., Ponater et al. (2012). This would be, however, computationally expensive. Therefore, we used the "fixed SST" method to quantify effective radiative forcing (ERF) as recommended by Forster et al. (2016).
We will include a short sentence in line 329, where we discuss the climate sensitivity.

**Old, l. 329** Under the reasonable assumption that the total radiative impacts (RIs) from the fSST experiments represent the corresponding ERFs with chemical rapid adjustments included (Winterstein et al., 2019), we calculate the climate sensitivity parameters $\lambda$ as $0.61 \pm 0.17$ K W$^{-1}$ m$^2$ and $0.72 \pm 0.07$ K W$^{-1}$ m$^2$, respectively.

[Figure]

Figure 1: Regression of surface temperature response against net radiative flux perturbation at the TOA for S5 MLO following Gregory et al. (2004).

**New, l. 329** The forcing strengths of $2\times$ and $5\times CH_4$ turn out too small to robustly quantify the corresponding climate sensitivity parameters $\lambda$ with a sensitivity analysis of the entire transient data following Gregory et al. (2004). Therefore, we calculate $\lambda$, under the reasonable assumption that the total RIs from the fSST experiments represent the corresponding ERFs with chemical rapid adjustments included (Winterstein et al., 2019), as $0.61 \pm 0.17$ K W$^{-1}$ m$^2$ and $0.72 \pm 0.07$ K W$^{-1}$ m$^2$, respectively.

> l. 99: I suppose methane is not an emission flux then? Would be good to clarify to avoid misunderstandings.

> Yes, that's right. The $CH_4$ mixing ratios are prescribed at the lower boundary.

We will add a clarifying sentence.

**Old, l. 92** Alike the fSST simulations, the $CH_4$ lower boundary mixing ratios of the mixed layer ocean (MLO) simulations are prescribed by Newtonian relaxation (i.e. nudging).

**New, l. 92** Alike the fSST simulations, the $CH_4$ lower boundary mixing ratios of the MLO simulations are prescribed by Newtonian relaxation (i.e. nudging). Thus, no $CH_4$ emission flux boundary was used, but pseudo surface fluxes were calculated by the MESSy submodel TNUDGE (Kerkweg et al., 2006) to reach the prescribed $CH_4$ lower boundary mixing ratios.

In addition, we will reformulate the following sentence.

**Old, l. 99** All other prescribed boundary conditions, such as emission fluxes, in the sensitivity simulations are identical to the respective reference simulations and represent conditions of the year 2010 in general.

**New, l. 99** Apart from $CH_4$, all other boundary conditions and emission fluxes used in the sensitivity simulations are identical to the reference simulations and represent conditions of the year 2010 in general.

> l. 139: One way of quantifying the importance of the climatological surface temperature differences would be to compare the global mean surface temperatures. I assume those differences should be smaller but possibly more relevant. Given that the MLO simulations are also free-running, could those effects

also just represent some form of internal variability, which, if I understand correctly could still affect the sea ice distribution through atmospheric variability and its effect on SSTs? Higher latitudes can show similarly large variability for fully coupled ocean models. Similar arguments could apply to the NH (cf. l. 143). Looking at Fig. S1, I would think that the overall difference is positive, but the visual effect overemphasizes those changes in SH high latitudes which make up quite a small area. For climate sensitivity aspects, I would actually be more interested in the differences in tropical low-cloud regions which appear to stand out?

You are right, the differences are smaller on the global scale than at higher latitudes. The highest differences occur near the sea ice edge, which poses the largest challenge to being reproduced by a thermodynamic ocean/ice model. While avoiding to let this section become too long, we have tried to improve the balance in the discussion of regional and global differences.

**Old, l. 136** The reduction of sea ice concentration (SIC) results in up to 1.5 K higher SSTs in the Southern Ocean in REF MLO compared to the prescribed climatology (see Fig. S1). Zonal mean air temperatures in the Southern Hemisphere (SH) extra-tropical troposphere are likewise up to 1 K higher in REF MLO compared to REF QFLX on annual average (not shown). As the contribution of Antarctic sea ice melting to global surface albedo feedback and climate response is comparatively small, a substantial underestimation of the climate sensitivity from this effect is not to be expected.

In the Northern Hemisphere (NH), the monthly climatology of sea ice area is generally well reproduced (see Fig. S2). However, in boreal winter and spring REF MLO overestimates the prescribed climatology of sea ice area with a maximum deviation of $1.33 \times 10^9$ km$^2$ in April. The larger SICs result in about 0.5 K lower SSTs on annual average in REF MLO in the Greenland Sea and in the Barents Sea (see Fig. S1), where the increase of SIC is located (not shown). In the Hudson Bay and in the Labrador Sea, on the other hand, the sea ice cover is reduced in REF MLO resulting in about 1 K higher SSTs in REF MLO compared to the prescribed climatology (see Fig. S1). The deviation from the prescribed climatology is strongest in this region in boreal summer. In summary, REF MLO simulates sufficiently realistic oceanic conditions for our purpose.

**New, l. 136** The reduction of SIC results in up to 1.5 K higher SSTs in the Southern Ocean in REF MLO compared to the prescribed climatology (see Fig. S1). In the NH, the annual cycle of the sea ice area is generally well reproduced (see Fig. S2), except for a slight overestimation of the sea ice area in REF MLO resulting in about 0.5 K lower annual mean SSTs in the Greenland Sea and in the Barents Sea (see Fig. S1). However, the sign of the global and annual mean surface temperature difference between REF MLO and REF fSST is determined by the positive REF MLO bias related to the Antarctic sea ice reduction. The global mean difference is 0.28 K, much less than the regional maxima near the ice edges, and with a small contribution of about 0.10 K from the tropical belt. It is unlikely that this will lead to substantial biases in the estimation of global mean surface temperature response and climate sensitivity in the intended equilibrium climate change simulations.

l. 174/175: "would be beyond the scope"?

Yes, thank you for this suggestion. We will reformulate the sentence.

l.180: It would indeed be useful to see the overall response, the rapid adjustment response and the difference due to slow feedbacks as subplots next to each other.

We understand that our previous presentation was difficult to follow, especially when not knowing the study of Winterstein et al. (2019). We decided to show 2x2 panel plots of the full response (MLO) and the slow feedbacks (difference between MLO and fSST) for S2 and S5 for temperature, OH, $H_2O$, and $O_3$. This should simplify the interpretation of the slow response. However, we decided to not show the rapid adjustments (fSST) again as this would duplicate the work of Winterstein et al. (2019). As the slow feedbacks impose only small modifications, the patterns of the full response and the rapid adjustments are qualitatively very similar and it should be possible to follow the presentation.

l. 193: the tropopause is defined how?

Here, we used a climatological tropopause calculated as:
$tp_{clim}= 300 \text{ hPa} − 215 \text{ hPa} \cdot \cos^2(\phi)$
The used troposphere definition is recommended by Lawrence et al. (2001), when calculating the $CH_4$ lifetime. We will add the following sentence to the text.

**Old, l. 192** B is the region, for which the lifetime should be calculated, e.g. all grid boxes below the tropopause for the mean tropospheric lifetime.

**New, l. 192** B is the region, for which the lifetime should be calculated, e.g. all grid boxes below the tropopause for the mean tropospheric lifetime. For the $CH_4$ lifetime calculation a climatological tropopause, defined as $tp_{clim}= 300 \text{ hPa} − 215 \text{ hPa} \cdot \cos^2(\phi)$, with $\phi$ being the latitude in degree north, is used as recommended by Lawrence et al. (2001).

l. 228: revise sentence

In response to a comment by referee 1, we have restructured the whole paragraph (see answer to referee 1 to reply to line 232).

l. 258: you mean 'stratospheric abundance'

Actually, we referred to the abundance of $H_2O$ in the troposphere and the stratosphere here. The abundance of tropospheric $H_2O$ is indirectly influenced by $CH_4$ through the $CH_4$-induced tropospheric warming. However, we admit that the first sentence was not very meaningful and we decided to restructure the paragraph.

**Old, l. 258** $H_2O$ is a precursor of OH and its abundance is also influenced by $CH_4$ mixing ratios. Winterstein et al. (2019) reported a steady increase of $H_2O$ with height for the $CH_4$ doubling and fivefolding experiments with prescribed SSTs and SICs. Figure 6 shows the difference of the $H_2O$ response between the MLO and the fSST simulations (see Fig. 5 in Winterstein et al., 2019 and Fig. S8 for the respective response patterns of $H_2O$ in the fSST and the MLO simulations, respectively). As the saturation vapour pressure increases with temperature, the warming of the troposphere in the MLO simulations consistently leads to a stronger increase of the tropospheric $H_2O$ mixing ratio in comparison with the respective fSST simulation. The maximum difference between MLO and fSST can be found in the upper tropical troposphere and extratropical lowermost stratosphere and reaches 11 percentage points (p.p.) (35 p.p.) for the $2\times$ ($5\times$) $CH_4$ experiments.

**New, l. 258** Winterstein et al. (2019) reported a steady increase of stratospheric water vapour (SWV) with height for the fSST experiments as an outcome of the enhanced $CH_4$ depletion as discussed in the previous

paragraph, whereas tropospheric $H_2O$ remained largely unaffected. The warming of the troposphere in the MLO simulations consistently leads to an increase of the $H_2O$ mixing ratios also in the troposphere as evident from Fig. 6. The maximum difference in tropospheric $H_2O$ response between MLO and fSST can be found in the upper tropical troposphere and extratropical lowermost stratosphere and reaches 11 p.p. (35 p.p.) for the $2\times$ ($5\times$) $CH_4$ experiments.

Figure 6: another case where it would be useful to see the overall response as well instead of just the difference to the rapid adjustment response. Same for Figure 7. 2x2 panels.

Yes, we agree. As stated in the answer to the previous remark to line 180, we will show 2x2 panel plots of the full response (MLO) and the slow feedbacks (difference between MLO and fSST) for S2 and S5 for temperature, OH, $H_2O$, and $O_3$.

l. 334: the efficacy of ERF methane of close to 1 appears surprising to me – see e.g. the 145 Hansen et al. Efficacy of climate forcings, Journal of Geophysical Research (2005).

Looking at Table 1 of Hansen et al. (2005) we find efficacy values between 1.05 and 1.08 under the effective radiative forcing framework (with 1.5xCO2, equivalent to a forcing of 2.38 Wm-2 as a reference). This may seem at odds with the most recent work of Richardson et al. (2019), who suggest a $CH_4$ efficacy value well below 1. However, in their work the reference is 2xCO2 (equivalent to about 4 Wm-2, while the 3xCH4 simulation runs with 1.2 Wm-2 only). This is a dangerous comparison as the climate sensitivity parameter tends to depend on the strength of the forcing. Compare, e.g., with Hansen et al. (2005)'s 1.25xCO2 and 2xCO2 runs, and it becomes obvious that 3xCH4 vs. 1.25xCO2 would probably make a more fair comparison. Many recent studies also show, how delicate the climate sensitivity parameter of $CO_2$ can depend on the forcing strength.
We think, however, that the main difference between previous work and our study is the inclusion of ozone and water vapor contributions to the methane forcing. Thus, in a chemistry-climate model, the "effective climate sensitivity of methane" will probably contain components from pure $CH_4$, pure $O_3$, and pure stratospheric $H_2O$. Hence, the finding of an efficacy close to 1 in our framework is indeed a surprise that deserves further investigation.

l. 368: how is this calculation of the effect on stratospheric temperatures done precisely? Could you provide more detail about the calculations? Are they expected to be robust in different regimes of the atmosphere, e.g. in the lowermost stratosphere vs the tropical upper stratosphere? What is "addst" in equation (2)?

We feel that from our previous formulation it was not clear that the stratospheric adjusted temperature response is the one shown in Fig. 9 and Fig. S11. We will formulate this clearer and use the abbreviation $\Delta T_{adj}$ already, when introducing the calculation of stratospheric adjusted temperatures. "addst" is the EMAC internal abbreviation for the adjusted stratospheric temperatures. We agree that the naming is not very intuitive and will replace it by "adj".
The calculation of the adjusted temperatures response is regime-independent. However, it is not meaningful if the radiatively induced temperature adjustment initiates dynamic processes whose effects on the temperature field are stronger than the radiatively induced changes. This would be the case in the troposphere. However, as the stratosphere is highly stable the radiatively induced temperature response dominates. This is the case in the lower as well as in the upper stratosphere.

**Old, l. 354** Following Winterstein et al. (2019) we calculate the stratospheric adjusted temperature response to changes in $CH_4$, tropospheric and stratospheric $H_2O$, and tropospheric and stratospheric $O_3$, as well as

their individual contributions for S2 MLO and S5 MLO (see Fig. S11 for simulation S2 MLO and Fig. 9 for simulation S5 MLO).

The difference of the adjusted stratospheric temperature response between S5 MLO and S5 fSST is shown in Fig. 10 (for S2 see Fig. S12).

**New, l. 354** Following Winterstein et al. (2019) we calculate the stratospheric adjusted temperature response $\Delta T_{adj}$ to changes in $CH_4$, tropospheric and stratospheric $H_2O$, and tropospheric and stratospheric $O_3$, as well as their individual contributions, for S2 MLO and S5 MLO (see Fig. S11 for simulation S2 MLO and Fig. 9 for simulation S5 MLO). $\Delta T_{adj}$ represents the temperature response induced by composition changes of radiatively active gases (Stuber et al., 2001).

The difference of $\Delta T_{adj}$ between S5 MLO and S5 fSST is shown in Fig. 10 (for S2 see Fig. S12).

**Old, l. 368** By calculating the difference between the total temperature response in the regular simulations and the sum of the individual contributions of $CH_4$, $H_2O$ and $O_3$ to the adjusted stratospheric temperatures, we attempt to identify the dynamical effect ($\Delta \tilde{T}_{dyn.}$) in the stratospheric temperature response as

$$\Delta \tilde{T}_{dyn.} = \Delta T(SX\text{-}REF) - \Delta T_{addst}(SX\text{*-}REF\text{*})$$

with X being either 2 or 5. A similar approach was, for example, used by Rosier and Shine (2000) and Schnadt et al. (2002) to distinguish between the radiative impact of trace gases and dynamical contributions to the total temperature response.

**New, l. 368** By calculating the difference between the total temperature response in the regular simulations $\Delta T$ and the sum of the individual contributions of $CH_4$, $H_2O$ and $O_3$ to the adjusted stratospheric temperatures ($\Delta T_{adj}^{total}$, see Fig. 9 a) and Fig. S11 a)), we attempt to identify the dynamical effect ($\Delta \tilde{T}_{dyn.}$) in the stratospheric temperature response as

$$\Delta \tilde{T}_{dyn.} = \Delta T(SX\text{-}REF) - \Delta T_{adj}^{total}(SX\text{*-}REF\text{*})$$

with X being either 2 or 5. A similar approach was, for example, used by Rosier and Shine (2000) and Schnadt et al. (2002) to distinguish between the radiative impact of trace gases and dynamical contributions to the total temperature response.